



**Socio-Hydrologic Modeling of the Dynamics of Cooperation in the Transboundary**
**Lancang-Mekong River**
You Lu [1], Fuqiang Tian [1], Liying Guo [1], Iolanda Borzi [2], Rupesh Patil [3], Jing Wei [1], Dengfeng
Liu [4], Yongping Wei [3], David J. Yu [5,6], and Murugesu Sivapalan [7,8]
[1] Department of Hydraulic Engineering, State Key Laboratory of Hydro-science and
Engineering, Tsinghua University, Beijing 100084, China
[2] Department of Engineering, University of Messina, Italy
[3] School of Earth and Environmental Sciences, University of Queensland, St. Lucia, QLD
4072, Australia.
[4] State Key Laboratory of Eco-hydraulics in Northwest Arid Region of China, Xi'an
University of Technology, China
[5] Lyles School of Civil Engineering, Purdue University, West Lafayette, IN, USA,
[6] Department of Political Science, Purdue University, West Lafayette, IN, USA
[7] Department of Geography and Geographic Information Science, University of Illinois at
Urbana-Champaign, Urbana, Illinois 61801, USA
[8] Department of Civil and Environmental Engineering, University of Illinois at Urbana-
Champaign, Urbana, Illinois 61801, USA
Correspondence to:
Fuqiang Tian, tianfq@mail.tsinghua.edu.cn



Submitted to *Hydrology and Earth System Sciences*
Special issue: Socio-hydrology and Transboundary Rivers





**Abstract**
The transboundary Lancang-Mekong River Basin has experienced dynamics of cooperation
over the past several decades, which is a common emergent response in transboundary human-
water systems. Downstream countries rely on Mekong River for fisheries, agriculture, etc.,
while upstream countries have been constructing dams to generate hydropower. The dam
construction and operation in upstream countries have changed the seasonality of streamflow
in downstream countries, affecting their economic benefits. More recently, cooperation between
upstream and downstream countries has been enhanced throughout the river basin. In this study,
we introduce a quantitative socio-hydrological model to simulate hydrological processes,
reservoir operations, economic benefits, policy feedbacks and therefore dynamics of
cooperation within the Lancang-Mekong River basin. The model reproduces the observed
dynamics of cooperation in the basin revealed by sentiment analysis of news articles.
Hydrological variability such as droughts and human activities associated with reservoir
operations affect dynamics of cooperation between the riparian countries, with importance
attached to indirect political benefits of upstream playing an important role in the enhancement
of cooperation. In this way, our study generated understanding of emergent cooperation
dynamics in this transboundary river basin, and the socio-hydrological model used here
provides a useful new framework to investigate and improve transboundary water management
elsewhere.
**Keywords:** transboundary river basins, socio-hydrology, cooperation, emergent dynamics,
mechanistic modeling



**Introduction**

As an important and complex issue, transboundary water management has attracted increasing
attention and efforts globally. Transboundary rivers refer to rivers shared by two or more
countries (Wolf et al., 1999), or two or more states within individual countries. There are over
310 transboundary rivers spanning over 150 countries, covering more than 40% of the world's
human population and land areas (UNEP, 2016;McCracken and Wolf, 2019). Transboundary
water management in a reciprocal manner is critical to ensuring regional cooperation and
sustainable development and to achieving water security, food security, energy security and
ecosystem security for the human populations residing within these river basins. From a human
perspective, rivers serve multiple functions such as water supply, irrigation, fishery, navigation,
hydropower generation, and provision of numerous other ecosystem services. These functions
can vary spatially within a river basin, and consequently, societal preferences for water use may
also differ in different locations, leading to possible disputes and conflicts between upstream
and downstream uses. Under these circumstances, cooperation among the various stakeholders
is necessary, which requires equitable and reciprocal benefit sharing, for humans to realize the
full potential of the services that rivers provide. Cooperation could take different forms (Sadoff
and Grey, 2005), and operating at different levels (Sadoff and Grey, 2002). Forms and levels of
cooperation can vary from unilateral actions and disputes, to collaboration, to joint action, and
to integrated and coordinated approaches (Sadoff and Grey, 2005). For example, information
sharing for flood and drought mitigation, reservoir operations adapted to the needs of both
upstream and downstream users, joint ownership of water-related infrastructure and institutions
for basin-wide cooperation are common forms of transboundary river cooperation.



Compared to water resources management in domestic river basins, transboundary river
management must deal with an additional complexity. The complexity arises from the fact that
in transboundary river basins the different preferences for water uses may be separated by
national or state boundaries. Under these circumstances, cooperation among stakeholders could
be inter-twined with other issues, or are limited by riparian relations, compounded by
institutional limitations (Wolf et al., 1999) and differing national economic and strategic
interests. When combined with obstacles towards enforcement (Müller et al., 2017;Espey and
Towfique, 2004), cooperation is much more difficult to achieve. The incompatible requirements
or demands of different riparian countries or states and the absence of institutional arrangements
to reconcile these differences may lead to sub-optimal outcomes for all stakeholders, leading to
conflicts that may be harder to manage (Petersen-Perlman et al., 2017).
Despite the challenges in transboundary river cooperation, there are examples of successful
cooperation and avoidance of conflict. Having experienced great losses due to environmental
pollution and flooding, the countries sharing Rhine river in Europe have cooperated
successfully to address shared goals of environmental protection and flood control over the last
several decades (Schultz, 2009). However, there are also examples of failures to cooperate.
Due to lack of unified and cooperative actions of the riparian countries, Amu Darya River and
Syr Darya River in Central Asia suffered over-exploitation of their water resources which
resulted in consequent disappearance of most water surface and ecological disaster in the once
thriving Aral Sea (Tian et al., 2019). Therefore, it is worth investigating the fundamental
question of what made the difference between examples of successful cooperation and those of
conflict and failure.



Over the last several years, researchers have spent considerable efforts to analyze and
understand the aforementioned question. Extant studies include empirical researches as well as
modeling efforts (De Stefano et al., 2017;De Bruyne and Fischhendler, 2013;Bernauer et al.,
2012;Beck et al., 2014). The International Water Events Database has collected cooperative and
conflictive water interactions over global transboundary river basins, and provides useful data
and frameworks for further statistical studies (De Stefano et al., 2010;Munia et al., 2016) and
detailed investigations in specific basins (Feng et al., 2019). Statistical methods or case studies
help to identify the broad factors affecting transboundary river cooperation. These have
included: natural conditions (e.g., hydrological scarcity and variability) (Dinar et al.,
2010;Dinar, 2009), political relations (Zeitoun and Mirumachi, 2008), power dynamics
(Zeitoun et al., 2011;Petersen-Perlman et al., 2017), institutional arrangements (Dinar, 2009),
and the relative levels of social and economic development (Song and Whittington, 2004).
Hydrological-economic models that involve hydrological simulation and benefit calculation
and allocation through benefit maximization or game theory (Li et al., 2019;Yu et al., 2019b)
are also common methods used to analyze the human-water interactions in transboundary river.
These modeling approaches have been applied to the Lancang-Mekong and the Nile river basins
(Cai et al., 2003;Ringler and Cai, 2006;Arjoon et al., 2016;Basheer et al., 2018).
Most of the studies highlighted above have viewed the cooperation in transboundary rivers in
a static way. However, a key aspect of cooperation by social actors (in this case riparian
countries or states, representing the humans living in these states) is that cooperation itself is
dynamic in nature, and transboundary river cooperation is also evolutionary. For example, in
the Colorado River Basin shared by USA and Mexico, industrialization and population growth



have increased the stress on surface and groundwater resources and on water quality. Ground
water depletion and water pollution contributed to tension between the two countries from the
1940s. Following protracted negotiations, several treaties were signed and institutions built,
with the result that the interactions between USA and Mexico have now become more
cooperative in recent years (Frisvold and Caswell, 2000). Globally, the cooperative tendency
reached its peak during 1971-1986, compared to the previous 1948-1970 and later 1987-1999
periods (Wolf et al., 2003). The relatively low cooperative tendency over the 1987-1999 period
is thought to continue to the 2000-2008 period (De Stefano et al., 2010). The focus of
transboundary water treaties, which symbolize cooperation, have been reported to shift from
exploitation of water resources to sustainable water management and framework setting, with
increased importance given to environmental health (Giordano et al., 2013). The approaches
used in studies to date do not accommodate the dynamic co-evolutionary nature of
transboundary cooperation and conflicts and are therefore not up to the task of seeking
mechanistic explanations for observed dynamics of cooperation in transboundary river basins.
In this study, we aim to address this knowledge gap by adopting a process-based, socio-
hydrologic framework to represent transboundary cooperation in the Lancang-Mekong river
basin, which involves China, Myanmar, Laos, Thailand, Cambodia and Vietnam as riparian
states. Using dynamic modeling to understand the mechanisms behind cooperative or
conflictive actions of riparian countries, not only in a specific river basin, but also similarities
and differences between basins, would help in elucidating key drivers that account for
differences in the cooperation level and its dynamics over time. Increased mechanistic
understanding will help increase the scope of cooperation and avoidance of conflict in the future,



and generate economic, social, and geopolitical benefits (Yu et al., 2019a;Sadoff and Grey,
2002), which are expressed as "beyond the river" benefits by Sadoff and Grey (2002). Enhanced
cooperation could lead to harmony in human-water relations generally and regionally, including
equitable and sustainable use of water. Conversely, the continuation of conflicts could result in
disordered water use, over-exploitation (Tian et al., 2019) and overall loss of amenities.
In approaching this aim, it is critical to capture the two-way feedbacks between the social
system and the transboundary river system. Entering the Anthropocene era, human society and
hydrological systems have become ever more tightly coupled, and in the long-term, co-
evolution of the resulting coupled, socio-hydrological system has been shown to result in
emergent dynamics and unintended consequences (Sivapalan and Bloschl, 2015). Examples
include decadal asymmetric dynamics of human water consumption in several large semi-arid
river basins in Asia (Tian et al., 2019), and the "pendulum swing" in agriculture water use and
human development in both Eastern and Western Australia (Kandasamy et al., 2014). Socio-
hydrology as a science explores the two-way feedbacks between human and water systems,
necessary to understand and mimic observed emergent dynamics (Sivapalan and Bloschl, 2015).
Driven by both natural and social forces, a transboundary river basin can also be viewed as a
coupled socio-hydrological system, now with a distinct spatial (upstream-downstream)
dynamics mediated by multiple riparian states. Observed patterns of cooperation and conflict
in a transboundary basin can then be seen as a special case of emergent dynamics that results
from interactions and feedbacks between the actions of water users or stakeholders in upstream
and downstream riparian states and the interplay of associated hydrological, economic, and
social, and geo-political processes (Di Baldassarre et al., 2019). Historical patterns of the





intensity or levels of cooperation between riparian states are key indicators that can be used as
targets of socio-hydrologic models developed with the aim of generating mechanistic
understanding of the co-evolutionary paths followed by transboundary river basin management.
In this study, we will present a coupled socio-hydrological model developed to simulate the
dynamics of conflict and cooperation in transboundary river systems, and its application to the
Lancang-Mekong river basin. The nature of shared use of the waters of the Lancang-Mekong
River has significantly evolved over the last 60 years through cycles of cooperation and conflict.
The socio-hydrological model developed here is used to mimic the mechanisms of cooperation
and conflict in this basin in a way to gain basic understanding that may be transferred to
transboundary river basins elsewhere.
The remainder of the paper is organized as follows. In Section 2, we will introduce the study
area and the history of observed dynamics of cooperation and conflict. Section 3 will present
the rationale and details of the socio-hydrological model, including the various modules and
governing equations describing the various subsystems, and how they are coupled in a way to
capture the dynamics of cooperation and conflict. Section 4 presents the simulation results and
a discussion and interpretation of the results, followed by, in Section 5, a summary of the main
conclusions and the understanding and insights gained from the study.
**1.    Study Area and Historical Timeline of Cooperation and Conflict Dynamics**
Lancang-Mekong River is an important transboundary river located in Southeast Asia. As
shown in Figure 1, it originates from the Tibetan Plateau in China, and over its entire length of
4900 km it passes through Myanmar, Laos, Thailand, Cambodia, and Vietnam (Wang et al.,
2017). The Lancang-Mekong river basin drains an area of 812,400 km$^2$ and supports the water





needs and livelihoods of over 65 million people (Ringler and Cai, 2006;MRC, 2018;You et al.,
2014). The annual average discharge of Lancang-Mekong River flowing into the South China
Sea is close to 475 billion m³/year (Campbell, 2016). The drainage area of the upstream part,
i.e., the Lancang River Basin in China, is 195,000 km², which accounts for 24% of the whole
basin area. The Mekong River Basin in Myanmar, Laos, Thailand, Cambodia and Vietnam
covers an area of around 600,000 km² (Li et al., 2017).
Starting from a relatively undeveloped basin in the 1950s, Lancang-Mekong River Basin has
experienced rapid economic growth in recent decades (MRC, 2010). Although they all have
many shared interests, different riparian countries within the Lancang-Mekong river basin
benefit from different river functions. For example, while all riparian countries have the need
to protect themselves from the negative impacts of floods and droughts and ensure the
sustainability of ecosystem, the upper riparian states of China and Laos have constructed and
plan to construct many dams, mainly for hydropower generation (Keskinen et al., 2012). For
the downstream states of Thailand, Cambodia and Vietnam, agriculture and fishery are the main
uses of the Mekong River. Irrigated agriculture is a major water consumer in the basin (MRC,
2018), and rice is the main staple crop (Campbell, 2016). In the lower Mekong region,
especially in Cambodia and Vietnam, fishery not only employs a large number of people, but
also sustains their protein demands (Campbell, 2016).
As an important and geopolitically sensitive region (Campbell, 2016), Lancang-Mekong River
Basin has experienced both conflict and cooperation since the end of World War II under the
impacts of changing geopolitical relationships, hydrological dynamics and socio-economic
conditions. With the sponsorship of the United Nations Agency ECAFE, the Committee for



Coordination of Investigations of the Lower Mekong Basin was initiated in 1957, and early
efforts included the setting up of comprehensive hydrological observations and the setting up
of regional plans for hydropower, flood control and irrigation (Campbell, 2016). However,
because of the withdrawal of Cambodia in 1977 due to political reasons, Thailand, Laos and
Vietnam initiated the Interim Committee for Coordination of Investigations of the Lower
Mekong, which took limited efforts towards regional cooperation. Until 1995, the four countries
of the lower Mekong were part of the Agreement on the Cooperation for the Sustainable
Development of the Mekong River Basin, through which they established the Mekong River
Commission (MRC). MRC was designed to enhance cooperation on water utilization and
management, socio-economic development and ecosystem conservation (MRC, 1995).
Although China signed an agreement on the provision of hydrological information on the
Lancang-Mekong River in 2002, the efforts of MRC were limited due to the absence of the
upstream states, namely China and Myanmar. Finally, the Lancang-Mekong Cooperation
Mechanism (LMC) was initiated in 2016 to include all of the six riparian countries and thus
enhance more comprehensive cooperation (Feng et al., 2019).
Specifically, cooperation in Lancang-Mekong River in the 21$^{st}$ century has been in the spotlight
because of rapid changes in climatic and hydrological conditions, intensified human activity
and geopolitical sensitivity of the region. Dam construction principally in the two upstream
countries, China and Laos, has continued over three decades. Since 2010, large hydropower
plants have been commissioned on the mainstream of Lancang-Mekong River (Han et al., 2019).
Reservoir operations in the upstream increase dry season runoff and reduce runoff peaks during
the flood season (Hoanh et al., 2010). The resulting changes in river flow were strongest in the



upper Chiang Saen station in Thailand and less marked in the lower station Kratie in Cambodia
(MRC, 2018). The resulting change of seasonality of river flows has a significant impact on the
benefits of different water uses (Pokhrel et al., 2018), for example, wetland ecosystem services
(Dudgeon, 2000) in Vietnam, and fish capture in the largest freshwater lake in Southeast Asian,
Tonle Sap (Kite, 2001) located in Cambodia. Correspondingly, due to the effects of upstream
dam operations for hydropower generation, the downstream countries faced concerns about
benefit losses. Here the loss indicates deviation from their maximum expected benefit instead
of absolute loss, because human values outcomes as gains and losses relative to a reference
level (Kahneman and Tversky, 1979). To obtain indirect political benefits, which is described
as "diplomatic returns" in Yu et al. (2019b), the upstream country China has worked to change
flow regulations of their reservoirs to satisfy the demands of the downstream countries and
achieve regional cooperation. One example of this is the emergency water release from China
in 2016 to alleviate the effects of a severe drought in the lower Mekong basin (Yu et al., 2019b).
This change of hydropower dam regulations in upstream countries can be regarded as an
example of a cooperative response.
Figure 2 summarizes the hydrological and anthropogenic events in Lancang-Mekong River
Basin. The upstream countries China and Laos have constructed or planned to construct dams
on the mainstream of Lancang-Mekong River. Two major reservoirs on the mainstream,
Xiaowan and Nuozhadu, went into production in 2010 and 2012 respectively. The filling and
operation of reservoir caused the alteration of hydrological regimes in the downstream, i.e.,
increase of runoff in the dry season and reduction in the flood season. Economic losses
compared to expected benefits caused by the change of hydrological seasonality and natural




droughts, led to concerns raised by downstream countries, and tension and conflict. However,
cooperation has been enhanced in recent years, exemplified by some cooperative actions of the
upstream country China, such as emergency water release during a period of drought. We will
use the socio-hydrological model to simulate the water-related events and the cooperation
dynamics, and provide mechanistic explanations based on socio-hydrologic interpretation of
the emergent dynamics.
**2. Model**
We will here introduce a transboundary river cooperation socio-hydrological model (TCSH
model) that will be used to simulate the dynamics of cooperation and conflict observed in
Lancang-Mekong River Basin. The causal loop presented in Figure 3 introduces the main
components of the model. It simulates the change of river flow seasonality caused by reservoir
operations, which causes loss compared to expected benefits to downstream countries in
different sectors. The loss compared to expected benefits leads to demands by the downstream
countries for more cooperation from upstream countries, to which the upstream countries
respond with changes to their reservoir operations. The modeled levels of cooperation, and the
resulting changes to reservoir operations, are determined by a balance between hydropower
losses and indirect gain of geopolitical benefits by the upstream countries.
As seen in Figure 3, the socio-hydrological model couples four main parts, i.e., hydrological
simulation, reservoir operation, economic benefit calculation, and policy feedback. A
distributed catchment hydrological model is used to model natural streamflow inputs to the
dams and is calibrated using observations at several stations along the Lancang-Mekong River
and its tributaries. With available reservoir information, the reservoir operation module





simulates two basic scenarios, i.e., maximizing upstream benefits versus maximizing
downstream benefits. The results of these two operational scenarios are weight averaged to
calculate actual water releases and reservoir storages. The economic benefit calculation module
estimates the economic benefits for both upstream and downstream countries covering
hydropower, irrigation and fishery sectors based on outcomes of the hydrological simulation
and reservoir operation modules. The fourth module simulates the policy feedbacks through the
estimation of economic benefits and operation weights through two key variables, i.e.,
cooperation demand of downstream countries and cooperation level of upstream countries.
Outcomes of sentiment analysis of newspaper articles are used to evaluate the modeled
cooperation demand. The calculation step length of the model is one month. Each of these
components of the model is discussed in detail in the following sections.
**3.1 Hydrological simulation**
We use the distributed hydrological model THREW to simulate natural runoff of mainstream
and tributaries without impacts of reservoir operations, i.e., $Q_n$ in Figure 3. The THREW model
has been applied to many river basins successfully, including rivers derived from mountainous
areas and consisting of snow and glacier melt, and large-scale basins (Tian et al., 2006;Tian et
al., 2008;Li et al., 2012;Mou et al., 2008). Based on the Representative Elementary Watershed
(REW) approach (Reggiani et al., 1998), the THREW model uses the REW as the sub-
catchment unit for hydrological simulations (He et al., 2015). The main runoff generation
processes include surface runoff, groundwater flow, and snow and glacier melt.
In this study, we divide the Lancang-Mekong basin into 651 REWs on the basis of DEM data,
as shown in Figure 1. The precipitation data is retrieved from TRMM data of 1998-2018. The



accuracy of TRMM data for hydrological simulation in this region has been proven successfully
(MRC, 2018). Thirty-two meteorological stations dispersed around the whole basin provide
meteorological inputs, including temperature, wind speed, humidity and radiation to calculate
potential evapotranspiration based on Penman-Monteith equation. Soil data is extracted from
FAO world soil database, and LAI, NDVI and snow are obtained from MODIS data. Daily
runoff observations of 6 stations on the mainstream of the Lancang-Mekong river include data
of Jinghong (1998-2013), Chiang Saen (1998-2015), Luang Prabang (1998-2015), Nong Khai
(1998-2007), Nakhon Phanom (1998-2015) and Pakse (1998-2006).
The hydrological model is used to provide simulations of natural runoff without the impacts of
water withdrawal and reservoir operations. Therefore, we use the runoff data in the period
before large reservoir construction for parameter calibration, i.e., runoff data of the period of
1998-2009. Runoff data of the hydrological stations on the mainstream are used for distributed
calibration, i.e., the parameters are calibrated separately and in a spatially distributed manner.
Specifically, the year of 1998 is used as a warm-up period, 1999-2004 as calibration period,
and 2005-2009 is set as validation period. The simulated runoff of 2000-2018 is used as natural
flow of mainstream tributaries $Q_n$ before the impacts of human activities.
**3.2 Reservoir operation**
Reservoir construction in the upstream reaches of the Lancang-Mekong River has accelerated
since 2000, and several large reservoirs on the mainstream have been constructed or are under
construction. Among them, the largest two reservoirs in China with seasonal runoff regulation
capacity (Yu et al., 2019b), namely Xiaowan and Nuozhadu, went into operation in 2010 and
2012 respectively. The basic information of Xiaowan and Nuozhadu including the total





reservoir storage $S_{total}$, dead reservoir storage, and flood limited storage $S_{flood}$ are listed in
Table 1. The total storage of the two major reservoirs account for 90% of the total storage of
the largest six reservoirs (Han et al., 2019). The cascade of reservoirs within China is simplified
and approximated in this study by the Xiaowan and Nuozhadu reservoirs.
Laos has aimed to be the "battery of Southeast Asia" (Stone, 2016) and has started hydroelectric
dam construction on the mainstream of the Mekong river in line with this ambition. Before that,
Laos constructed many dams on its tributaries, which also impact the streamflow regimes of
the Mekong River. According to MRC (2018), the expected live storage of reservoirs in Laos
will ultimately reach 24,257 MCM, accounting for 73% of the flows left for the four
downstream countries. For simplicity, we only consider the completed tributary reservoirs in
Laos. They are aggregated by one proxy reservoir in the upper reaches, including some
reservoir storages located in the relatively lower reaches in Laos (Li et al., 2019;WLE, 2018).
In the model, the proxy reservoir used is assumed to have live storage from 5,074 MCM in
2000 to 21,066 MCM in 2018, which was linear interpolated and represents continuous dam
construction in Laos.
Overall, these simplifications through lumping the effects of many reservoirs is deemed
reasonable for the purposes of this study, because three reservoirs (Xiaowan and Nuozhadu in
China and the aggregated Laos Reservoir) shown in Figure 4 capture most of the effects of
reservoirs within the entire river basin. As shown in Figure 4, the river system and its water
diversion configuration are also simplified, where T0, T1 to T6 indicate natural runoff of
upstream and tributaries, W4, W5, W6 are the water withdrawal for irrigation in Laos, Thailand,
Cambodia and Vietnam. For each node, runoff flowing to the next node is calculated by water



balance equation, e.g., for Thailand,
$$QN4 = QN3 + T5 - W4 \tag{1}$$

where, QN3 is runoff flowing to Thailand from the upstream node, Laos, T5 is inflow from
tributaries in Thailand, W4 is irrigation withdrawal in Thailand, and QN4 is runoff flowing to
the downstream node, Cambodia.
For the operation of constructed dams, we consider two basic scenarios. The first scenario is
the self-interested scenario (non-cooperation scenario, abbreviated by NC), in which the
upstream countries, China and Laos, operate the dams considering only their own hydropower
benefits $B_h$.
$$B_h = \text{ph} \times 9.81 \times Q_r \times \Delta\text{h} \times \eta \tag{2}$$

where, ph is the electricity price, $Q_r$ is the monthly water release from the reservoir, $\Delta$h is
the water head difference between the upstream and downstream, which is related to the actual
storage $S_r$, and $\eta$ is hydropower generation efficiency.
$$Q_{r,t} = max\{S_{r,t-1} + Q_{in,t} - S_{total}, 0, Q_{eco}\}, \text{ t} = 1,2,3,4,5,11,12 \tag{3}$$

$$Q_{r,t} = max\{S_{r,t-1} + Q_{in,t} - S_{flood}, 0, Q_{eco}\}, \text{ t} = 6,7,8,9,10 \tag{4}$$

Under this scenario, dams keep at their total storage $S_{total}$ during the dry season (November
to May) and their flood limited storage $S_{flood}$ in flood season (June to October). If the actual
storage of t-1 period $S_{r,t-1}$ is less than these two values the reservoir will store water to reach
the amount; otherwise, the reservoir will release water. There are also constraints of minimum
ecological release flow $Q_{eco}$ to satisfy the requirements of ecosystem and navigation. Actual
water release under the self-interested scenario $Q_{r,NC}$ is calculated using Equations (3) and (4).
The actual storage of next month $S_{r,t}$ is calculated based on water balance equation.


The second scenario is the altruistic scenario (full-cooperation, abbreviated by FC), where the
upstream countries operate dams to maximize the benefits of downstream countries. The
calculation of the benefits to downstream countries will be introduced in Section 3.3. Under
this scenario, the constraints contain maximum storage during dry season, maximum storage
during flood season, minimum storage of dead storage and minimum ecological release flow.
Then the processed results of actual water release $Q_{r,FC}$ will be used to calculate actual
reservoir storage $S_r$ based on the water balance equation.
As shown in Figure 3, with the calculated water release under the self-interested scenario $Q_{r,NC}$
and that under the altruistic scenario $Q_{r,FC}$, we obtain the weighted average scenario (WA
scenario) and final actual water release $Q_r$ by calculating their weighted average.

$$Q_r = Q_{r,NC} \times \delta_1 + Q_{r,FC} \times \delta_2 \tag{5}$$

where $\delta_1 + \delta_2 = 1$, and $\delta_2$ is calculated using the cooperation equations while $\delta_1$ is
calculated as the residual $1 - \delta_2$, which will be introduced in section 3.4. It should be noted
that the calculated $Q_r$ by equation (5) should be revised if it violates the constraints of
maximum storage during dry and flood seasons, minimum storage of dead storage and
minimum ecological release flow. And the final actual reservoir storage $S_r$ is calculated for
hydropower benefit calculation.
**3.3 Economic benefit calculation**
In this study, we consider the hydropower benefits $B_h$ of China and Laos, and agriculture
benefits $B_a$ and fishery benefits $B_f$ of Thailand, Cambodia and Vietnam. The hydropower
benefits calculation of China and Laos were introduced in section 3.2. Here agriculture benefits
$B_a$ only include irrigated rice without consideration of rain-fed crop production. Agricultural





water withdrawals dominate water consumption in the downstream countries, and rice is the
staple crop in this area. In this study, we use the FAO 33 crop water production function to
calculate crop yields and irrigation benefits (Doorenbos and Kassam, 1979).
$$B_a = pa \times Y_a \times A \tag{6}$$

$$(1 - \frac{Y_a}{Y_m}) = K_y \times (1 - \frac{\text{AET}}{\text{PET}}) \tag{7}$$

where $pa$ is price of rice, $A$ is the rice irrigation area, $Y_a$ and $Y_m$ are actual and maximum
crop yields, respectively. $K_y$ is crop yield response factor, and AET and PET are actual and
maximum evapotranspiration respectively. The information on the price of rice, irrigation area,
rice yield and irrigation withdrawal of Thailand, Cambodia and Vietnam are listed in Table 2.
$Y_m$ is set as 8.5 ton/ha for all three countries (FAO, 2004). AET and PET are calculated based
on potential evapotranspiration and irrigation amount, and the detailed methods could be found
in Allen et al. (1998) and Kaboosi and Kaveh (2012).
Fishery is one of the dominant environmental water uses in the lower Lancang-Mekong River
Basin, but it is difficult to quantify fishery benefits. In general, comprehensive fisheries models
have many required inputs to calculate fishery benefits, such as mortality, recruitment, and
fishing efforts (Baran and Cain, 2001). There are many studies focusing on the simulations of
fishery benefits through their relationships with water level (Hortle et al., 2005) and flooded
areas (Burbano et al., 2020). It is difficult to couple complex fishery models to our model, and
there is not any standard function for fishery benefits up till now. Here, for simplicity, we only
capture fishery benefits and do not include aquaculture benefits, since it is not significantly
impacted by hydropower operation. Based on literature review, an increasing function of runoff
with decreasing marginal increase was adopted to calculate capture fishery benefits, which is
simple but effective in Mekong Basin (Ringler, 2001;Ringler and Cai, 2006).
$$iff = \arctan\left(\frac{Q-Q_{min}}{Q_{max}}\right) \times \left(1 - b \times \left(\frac{Q-Q_{min}}{Q_{max}} - c\right)^2\right) \qquad (8)$$
$$B_f = pf \times iff - \text{Fcost} \qquad (9)$$
where $iff$ is the fishery production related to actual discharge $Q$, minimum discharge $Q_{min}$,
maximum discharge $Q_{max}$, and two parameters $b$ and $c$. In equation (9) to calculate fishery
benefit $B_f$, $pf$ is the fishery price and Fcost is fixed fishery cost. Overall, fishery benefits
for downstream countries are related to actual runoff, maximum runoff, and minimum runoff.
As shown in Figure 4, QN4, QN5, QN6 are used as actual runoff to calculate fishery benefits
for Thailand, Cambodia and Vietnam respectively.
**3.4 Policy feedback**
Cooperation demands $U$ of downstream countries arise from economic losses compared to
expected benefits, and the upstream countries take cooperative action to obtain indirect political
benefits, although this might reduce their hydropower generation benefits. Cooperative actions
of upstream countries take effect in multiple forms, such as information sharing and joint
investment (Sadoff and Grey, 2002). It is always difficult to quantify cooperation demand and
cooperation level. As a first attempt, in this study we only consider change of operation rules
of reservoirs as cooperative action and define the cooperation level $C$ of upstream countries
as the weight assigned to the operation rules to maximize downstream benefits when upstream
countries operate their reservoirs, i.e., $\delta_2$ in section 3.2. When cooperation level $C = 1$,
upstream countries operate dams to maximize the downstream benefits, i.e., altruistic scenario.
If $C = 0$, upstream countries will follow operation rules in Equations (3) and (4), which is
consistent with the self-interested scenario.



Following the assumption that cooperation demand is increased due to economic losses
compared to reference level, larger economic losses will cause greater community concerns and
thus increased cooperation demands. According to the theory of reference dependence, humans
evaluate gains and losses relative to a reference point (Schmidt, 2003), and the reference point
could be the status quo (Tversky and Kahneman, 1991) or the level of aspiration (Siegel, 1957).
Here we value the losses relative to the expected maximum benefits of sectors $B_{amax}$ and
$B_{fmax}$, i.e., as the differences between expected maximum benefits and actual benefits. As
shown in equation (10), we assume that the cooperation demand is proportional to economic
losses, but the sensitivity of each economic sector is distinct.

$$U = \varepsilon_a \times \frac{B_{amax} - B_a}{B_{amax}} + \varepsilon_f \times \frac{B_{fmax} - B_f}{B_{fmax}} \qquad (10)$$

where $\varepsilon_a$ and $\varepsilon_f$ are the sensitivity of agriculture loss and fishery loss. The sensitivities
indicate the importance of each sector to the overall lower basin economy, and larger sensitivity
means that downstream countries are more sensitive to the sector benefit change, and the unit
sector loss could lead to severer negative impacts. The expected maximum benefits $B_{amax}$
and $B_{fmax}$ are also used for normalization.
For the cooperation level of upstream countries, we use a logit dynamics model (McFadden,
1981;Hofbauer and Sigmund, 2003) taken from environmental economics. This model is used
to relate economic losses and benefits with the probability of cooperation. It has been widely
used and proven effective to relate natural system dynamics with cooperation dynamics, e.g.,
the simulations of cooperation on pollution control among stakeholders, who behave
responding to other stakeholders' behaviors and their own benefits (Iwasa et al., 2007;Suzuki
and Iwasa, 2009a, b). In the logit dynamics model, the probability of cooperation Pr could be





calculated as below:

$$\mathrm{Pr} = \frac{e^{\beta \times B_C}}{e^{\beta \times B_C} + e^{\beta \times B_N}} \tag{12}$$

where $\beta$ is a shape parameter ranging from 0 to 1, $B_C$ is the benefit of cooperation, and $B_N$
is the benefit without cooperation.
Similarly, for upstream countries, if they choose not to cooperate, their benefit $B_N$ will be
hydropower generation benefits under self-interested scenario $B_{h,NC}$ and benefits from other
sectors. If they choose to cooperate, besides the hydropower benefits under the altruistic
scenario $B_{h,FC}$ and benefits of other sectors, the upstream country will also gain indirect
political benefits, which is related to the cooperation demands of downstream countries. Here
we assume that the political benefit is proportional to cooperation demand $U$ and a political
factor $P$. If the upstream country values the political relations with downstream countries and
regards diplomatic benefits as important, as China has demonstrated in recent years, the value
of political factor $P$ will be higher. Therefore, the equation to calculate the actual cooperation
level $C$ for China is as described below, and the cooperation level for Laos should consider
agriculture benefits additionally.

$$\frac{\mathrm{dC}}{dt} = s \times \left[ \frac{e^{\beta \times (U \times P + \varepsilon_h \times \frac{B_{h,FC}}{B_{hmax}})}}{e^{\beta \times (U \times P + \varepsilon_h \times \frac{B_{h,FC}}{B_{hmax}})} + e^{\beta \times \varepsilon_h \times \frac{B_{h,NC}}{B_{hmax}}}} - C \right] \tag{13}$$

where $s$ is the responsive change rate reflecting the response speed of upstream countries, $\varepsilon_h$
is the sensitivity of hydropower loss, and $\frac{\mathrm{dC}}{dt}$ indicates the change of cooperation level
compared to the last period. As mentioned before, the cooperation level $C$ equals the weight
$\delta_2$, so the cooperation demand and cooperation level will affect reservoir regulations, and in
this way will drive the co-evolution of the coupled transboundary socio-hydrological system.
Empirical observational data is needed to evaluate the simulation of policy feedbacks. It is



difficult to measure cooperation demand, particularly the cooperation among countries on a
specific item, i.e., reservoir operation and water resources management. Sentiment analysis is
an emerging tool to quantify social data, which exploits the denotation of words and assigns
sentimental value to text strings by an algorithm (Bravo-Marquez et al., 2014;Abdul et al.,
2018). It has already been used to provide information of the attitudes of Chinese citizens
towards dam construction (Jiang et al., 2016). In this study we use the method of sentiment
analysis of newspaper articles in Thailand, which are assumed to reflect the changes in
cooperation demands of downstream countries. Newspaper articles could reflect public opinion
on issues of interest to the community, which have been used in previous socio-hydrologic
studies to monitor the evolution of environmental awareness vis a vis economic livelihood (Wei
et al., 2017).
We used the Lexis-Nexis database to extract relevant information in English newspapers in
Thailand (Weaver and Bimber, 2008), sorted the data manually and conducted sentiment
analysis. Although the English newspapers could omit some information when compared to
local language newspapers, they are important sources to analyze the dynamics of local public
opinions. Firstly, key words for search (e.g., Mekong, water, dam, etc.) and search limitations
(e.g., location of publisher) are set for this study, and data retrieval is conducted automatically.
Secondly, manual data sorting was used to remove duplicates and irrelevant news. Thirdly, the
sorted data was analyzed through coding to get the sentiment of each piece of news and
corrected manually. This method has been used widely to explore the perspectives towards
specific topics and the detailed steps have been introduced in Wei et al. (2020). Finally, each
piece of valid data will provide information of news titles, publication year, sentiment category



(positive or negative) and sentiment values. The sentiment values range from -1 to 1, with
positive values indicating positive sentiment of the news towards the topic. We will use the
sentiment values to evaluate simulated cooperation demand of downstream countries.
**3.  Results**
**3.1 Hydrological simulation and reservoir operation**
As major mainstream dams commissioned after 2010, the runoff data before that time could
roughly represent the natural runoff in Lancang-Mekong River. We use the observed runoff data
of 1999-2005 at Jinghong, Chiang Saen, Luang Prabang and Pakse for parameter calibration of
hydrological model, and use the rest of the data for validation. As shown in Figure 5, the
simulations at the four stations perform well with NSEs above 0.8 for the calibration period.
The NSEs of validation period at Jinghong, Chiang Saen, Luang Prabang and Pakse are 0.83,
0.80, 0.79 and 0.87 respectively. For most years, the simulations of troughs during dry seasons
and peaks during flood seasons are reproduced rather well, except for some extreme flood
events when simulations underestimated the flow. The accuracy over the dry and flood seasons
is important for later simulation of water availability and economic benefits. Besides, we also
use observations at two other stations, Nong Khai (1998-2007), and Nakhonphanom (1998-
2009), for validation of the hydrological model. The NSEs at these three stations reach to 0.81
and 0.75 respectively, which indicates the applicability of the THREW model at different
locations across the Lancang-Mekong river basin.
According to the observations and simulations, the annual discharge from China to downstream
countries at Jinghong station (QL3 in Figure 4) accounts for 66% of the discharge at Chiang
Saen (QN2 in Figure 4) and 20% of the discharge at Pakse (QN4 in Figure 4). As simplified in





Figure 4, runoff observed in Laos and Thailand account for 23% and 57% of the discharge at
Pakse. The proportions of China and Laos in Pakse runoff are higher during non-flood seasons
(November to May), and the change of seasonality of discharge in China and Laos caused by
reservoir operations could affect the discharge and thus economic benefits in downstream
countries.
Two basic scenarios of reservoir operations were set up. The first basic scenario is the self-
interested scenario, when upstream China and Laos operate reservoirs following their own
operation rules guided by self-interest only, as introduced in section 3.2. The other basic
scenario is the altruistic scenario, when upstream countries China and Laos operate reservoirs
to maximize the benefits of downstream three countries. Based the two basic scenarios, the
weighted average scenario (WA scenario) is also analyzed. Water release from Xiaowan,
Nuozhadu and the proxy Laos reservoir vary under the three scenarios, and we compare them
with natural water release without reservoir operation (NR scenario) in non-flood seasons. We
set the initial reservoir storage to maximum storage at the beginning of the year and simulate
the water release under two natural hydrological conditions, i.e., dry year of 2015 and normal
year of 2017. Initial value of cooperation level of China and Laos are both set to 0.5.
As shown in Figure 6(a-c and g-i), for both dry and normal years, the NC scenario keeps the
largest storages and the FC scenario keeps the lowest storages. In a dry year like 2015, with the
same cooperation level as in the normal year of 2017, reservoir storages under FC and WA
scenarios are lower to satisfy the demands of downstream countries. Water releases from the
three reservoirs under different scenarios in non-flood seasons in 2015 and 2017 are shown in
Figure 7. The final weighted average water releases (WA scenario) from Nuozhadu and Laos



Reservoirs to downstream countries are higher than natural water releases (NR scenario) during
non-flood season (December to May), especially in the dry year of 2015. It is consistent with
the phenomenon that reservoir operations increase discharge during non-flood seasons in
downstream countries in recent years.
As shown in Figure 8, the simulated reservoir storages under continuous WA scenario are lower
than the simulated storages under continuous NC scenario in all three reservoirs. As a
cooperative action, reservoir regulations under the continuous WA scenario keep releasing more
water, particularly during dry years when the demands of downstream countries are high. The
simulated storage of Xiaowan and Nuozhadu under continuous WA scenario keep a relatively
low level, because China values the indirect political benefits from downstream countries,
which leads to high cooperation level of China, as it will be introduced in detail in section 4.3.
**3.2 Economic benefit**
China and Laos operate reservoirs to obtain hydropower benefits, and the agriculture benefits
of Laos are also calculated. For each of the three downstream countries, benefits of agriculture,
fishery and wetlands are simulated individually. Overall, the economic benefits simulations
under WA scenario in each country and sector are reasonable compared to statistical data, as
listed in Table 3.
Under the continuous WA scenario, China and Laos have obtained increasing benefits mainly
due to ongoing dam construction. As Figure 9 shows, the simulated hydropower benefits of
China approached 1,800 million USD in 2018, which is reasonable since the annual generation
of the two reservoirs is close to 40 billion kWh (Yu et al., 2019b). The Laos reservoir generated
hydropower around 1,100 million USD while the statistical estimation of hydropower benefit





to Laos in 2015 is 1,076 million USD (MRC, 2018), proving the validity of economic benefit
simulations in Laos. In Figure 9(a), the hydropower benefit of China under WA scenario is
lower than NC scenario and higher than the FC scenario after 2012, indicating that cooperation
actions (WA and FC) could harm the hydropower benefit of China. It is similar in Laos, as
shown in Figure 9(b), but the benefits under WA resemble NC scenario are more due to the low
cooperation level of Laos. The differences between the blue and red lines indicate the losses
China and Laos need to bear if they cooperate altruistically to satisfy downstream demands and
maximize downstream benefits.
When the two major reservoirs in China went into operation and cooperation levels increased
after 2012, the total benefits of downstream three countries under WA scenario are higher than
the NC scenario, although they cannot reach the high level of the FC scenario when China and
Laos operate reservoirs merely for downstream benefits, as shown in Figure 10(a). The increase
of downstream benefits under WA scenario is remarkable compared to NC scenario (e.g., 420
million USD in 2018), indicating the significance of cooperation of upstream countries for the
benefits in downstream countries. The losses China and Laos need to bear is less than the gain
of downstream countries, which help to rationalize cooperation actions to enhance regional
benefits and is consistent with outcomes of simulations in other studies (Yu et al., 2019b;Li et
al., 2019;Do et al., 2020). Notably, in the dry years of 2015-2016, cooperative action of
upstream countries could mitigate the losses of downstream countries, but their benefits would
still be lower compared to those in normal years.
The downstream benefits of agriculture and fishery under the WA scenario are shown in Figure
10(b). Simulated agriculture benefit in 2018 is around 4,000 million USD with irrigation





withdrawals of 35 billion m$^3$, while the statistical irrigation withdrawal of the three countries is
47 billion m$^3$ (FAO, 2019). The simulated agriculture benefits of Thailand, Cambodia and
Vietnam are 1,355, 595 and 2,011 million USD respectively, which are consistent with the
statistical values for irrigated rice in Table 4, i.e., 1,314, 592 and 2,727 million USD. Statistical
values for irrigated rice are calculated by the irrigation areas (Cramb, 2020), irrigated rice
production per unit area and rice price (MRC, 2018).
As for the capture fishery benefits, the losses during the years of reservoir filling and droughts
are remarkable, approaching 224 and 181 million USD in 2010 and 2015, respectively. The
reduction of fishery capture is consistent with the outcomes of study by Orr et al. (2012), which
estimated that losses of fishery capture could reach to 20% with the impacts of upstream dams.
In 2018, the simulated fishery benefits of Thailand, Cambodia, Vietnam and the total fishery
benefit are 116, 1,146, 178, and 1,440 million USD, while the corresponding statistical values
are 120, 1,188, 195 and 1,503 million USD. The statistical fishery values are estimated on the
basis of fishery production (Burbano et al., 2020) and fishery prices (MRC, 2018). Overall, the
simulated benefits of downstream countries in the three economic sectors are basically
consistent with statistical values.
**3.3 Cooperation demand and level**
As introduced in section 3.4, two key variables in the policy feedbacks contain cooperation
demand of downstream countries and the actual cooperation level of upstream countries. In the
model, cooperation demand of downstream countries was assumed to be related to the losses
in the different sectors compared to maximum possible benefits, and the sensitivity to
agriculture loss and fishery loss, expressed in terms the parameters $\varepsilon_a$ and $\varepsilon_f$. We calculated



the cooperation demands of the three downstream countries based on benefit calculations, and
the simulated cooperation demands reached to high levels in 2004-2005, 2010, 2012-2016
(Figure 11). These peaks are caused by benefit losses compared to other years. The losses in
2004-2005 and 2015-2016 arose from recorded droughts (MRC, 2018), while the losses in 2010
and 2012-2014 are related to the constructions and operations of Xiaowan and Nuozhadu dam.
Cooperation levels of China and Laos are simulated separately so that they could decide their
own weights used in reservoir operations. Cooperation levels are related to downstream
cooperation demand $U$, political factor $P$ reflecting how much upstream countries value the
indirect political benefits that can be gained from downstream countries, upstream benefits
when cooperating or not, and the change rate $s$ that reflecting the response speed of upstream
countries. Compared to Laos, China regards the geopolitical values and diplomatic relations as
more important (Urban et al., 2018). Therefore, the political factor $P$ and change rate $s$ of China
are set as 2 and 0.1, respectively, while those of Laos are 1 and 0.02, respectively. As shown in
Figure 11, the cooperation level of Laos increased from the start at a slow speed and exceeded
0.17 in 2018. The slowly increasing trend of cooperation level of Laos could be reflected by
the on-going disputes and negotiations between Laos and other MRC members in respect of
reservoir construction by Laos on the mainstream of Mekong River since 2009 (Hensengerth,
2015). China finished the first major dam construction in 2010 and the increase of cooperation
levels started then. Compared to Laos, the increase rate is much higher, especially when the
major reservoirs were constructed and China adjusted their operational rules before 2015. The
rapid increase of cooperation level of China could be proven by the cooperative actions from
China in recent years. China initiated Lancang-Mekong Cooperation (LMC) framework in 2015,



which is a much broader framework that goes beyond water cooperation. When the historically
severe drought hit Mekong Basin in 2015 and 2016, China implemented emergency water
release to mitigate the negative impacts of droughts in downstream (Middleton and Allouche,

624   2016).

To evaluate the simulation outcomes of cooperation demands of downstream, we conducted
sentiment analysis towards the reservoir constructions and operations in upstream China and
Laos. Because the analyzed newspaper needs to be in English due to the language difficulty,
and we can only obtain continuous and relevant English newspapers in Thailand among the
downstream countries, we selected Thailand newspaper articles for the sentiment analyses used
for evaluation. The data processing is similar with that used in Wei et al. (2020), but we adjusted
the key words and filtering rules to fit our goals. From the database of Lexis-Nexis, we extracted
in total 4,622 pieces of data with keywords related to the dam constructions and regulations in
China and Laos, which are published by Thai newspapers. Then we selected 592 pieces of
relevant articles by removing duplicates and irrelevant news manually. The 592 valid pieces of
news cover the period of 2000-2018. Through coding and manually correcting, the sentiment
values of each piece of news are provided for statistical analysis. As shown in Figure 12, the
number of news articles concerning the impacts of upstream reservoirs increased significantly
after 2010, from less than 20 pieces each year to over 70 pieces in recent years. The means of
sentiment values fluctuate greatly in early years, which is caused by relatively small numbers
of pieces of news. In 2004, 2010-2012 and 2015, sentiment results reached low values through
the years, reflecting that the concerns and criticisms from Thailand towards China and Laos on
dam operation were high compared to normal years. The dynamics of sentiment values are



basically consistent with the simulations of cooperation demand of Thailand shown in Figure
12. Simulated cooperation demands of Thailand are high during 2005, 2009-2010, 2012-2015.
Similar to the cooperation demand of the three downstream countries introduced before, the
peaks of cooperation demand and concerns from downstream in 2005 and 2015 are ascribed to
droughts and losses, while the concerns in 2010 and 2012 are due to the effects of dam
constructions at Xiaowan and Nuozhadu during these two years. According to Wei et al. (2020),
topic analysis shows that most of the negative publications in Mekong countries are related to
the constructions and operations of dams, which is consistent with our results. Besides the
factors mentioned above, based on the text information of news, another reason why concerns
increased in 2010-2012 is that Laos started to construct Xayaburi dam, which is the first dam
Laos constructed on the mainstream of Mekong River and is regarded as a violation of the 1995
Mekong Agreement (Herbertson, 2013). Overall, our simulations of cooperation demands
reflect the empirical dynamics of downstream countries obtained through sentiment analyses.
**4. Discussion and Conclusions**
This paper presented the development and application of a coupled socio-hydrological model
to simulate the dynamics of cooperation and conflict in the transboundary Lancang-Mekong
river basin in Southeast Asia. Lancang-Mekong is a typical transboundary river where the
upstream mountainous area is rich in hydropower and lower plain areas are suitable for
irrigation and are rich in fisheries. Dam construction and operations in upstream countries
(China, Laos) have changed the seasonality of downstream river flows, which have impacted
the benefits gained by downstream countries, notably in terms of agriculture and fishery, both
of which rely on the discharge of rivers. When downstream countries faced benefit losses



compared to maximum benefits as a result, they led to community concerns, which they tend
to blame on upstream countries. Once the dams were constructed and were in place, the most
available and effective cooperative action to avoid regional conflicts was to operate the
reservoirs in a way to achieve basin-wide synergy between upstream and downstream countries
(Do et al., 2020). While upstream countries may have lost some economic benefits by
sacrificing some of their hydropower generation to benefit downstream countries, by doing so
they also stood to gain indirect political and economic benefits, e.g., better diplomatic relations
and more investment opportunities in downstream countries (Sadoff and Grey, 2002).
The socio-hydrological model presented in this paper captured the dynamics of such
cooperation and conflict through the coupling of modules representing hydrology, reservoir
operation, economic benefits and policy, which is simple but comprehensive. The interplay
among hydrological, economic and political factors is important, because hydrological
variability and human activities could impact the dynamics of cooperation jointly. The model
simulations were evaluated by using empirical observations of runoff and published statistics
of economic benefits in the different sectors. The model simulated cooperation demands by
downstream countries reached to high levels during dry years of 2004-2005 and 2015-2016,
and the dam filling years of 2010 and 2012. These patterns were consistent with outcomes of
sentiment analysis carried out based on articles published in English language newspapers in
one downstream country, Thailand, proving the validity of policy feedbacks embedded in the
socio-hydrological model.
A novel feature of the model is the quantification of policy feedbacks between upstream and
downstream countries in the form of a logit dynamics model. The logit dynamics model





operates in a way that willingness to cooperate increases when there are greater benefits to be
gained if the parties cooperated and fewer benefits if they do not. A particular strength of the
logit model is that it could explicitly include geopolitical factors that add to the indirect benefits
that upstream countries may gain through increased cooperation. The potential benefit increase
for upstream countries through cooperation, which may include direct economic benefits such
as eco-compensation and indirect political benefits, is assumed to boost their willingness to
cooperate. When upstream countries value the indirect political benefits more and are thus more
responsive to the downstream concerns, the cooperation level would increase, which is
consistent with the cooperative actions taken by China in recent years. Over the last two decades,
cooperation demands of the downstream countries increased over drought years and over the
years of reservoir filling. The surge of downstream concerns towards upstream countries needs
to be treated appropriately, otherwise the concerns could turn into more severe conflicts. The
losses of the downstream relative to maximum expected benefits could be mitigated by
cooperative actions of upstream countries, i.e., change of reservoir regulation, which will lead
to less concerns and less criticism from downstream countries.
As an early version transboundary river socio-hydrological model, there is significant room for
further improvement in the model formulation. The current model simulated the effect of
hydroelectric power generation in multiple dams in China and Laos in a lumped manner, which
has a negative impact on the accuracy of reservoir releases, and hence on benefit calculation
for downstream countries. The situation can be improved in the future through more distributed
simulation of a cascade of reservoirs. Additionally, in order to integrate the complex hydro-
economic relationships into the model, agriculture and fishery benefits are calculated in the



present model with rather simplified equations. There is room for significant improvement in
these benefit calculations. Flood control is one of the important functions of existing and
planned future dams, but has been ignored in this study, which may have under-estimated
benefits to both upstream and downstream countries. In future studies, with inclusion of more
accurate reservoir operation rules, hydro-economic relationships and consideration of flood
losses and impacts on ecosystem, a more advanced model could be used for sensitivity and
scenario analyses under future scenarios of possible climatic, socio-economic and political
changes. Sensitivity analysis will help to identify dominant influential factors and explore the
consequences of changes to the coupled human-nature system and upstream-downstream
feedbacks. Climate change and the expansion of human activities, including reservoir
construction and irrigation area expansion, will affect the water supply, water demand,
economic benefits and cooperation dynamics in transboundary rivers. Simulations under
different scenarios of climate change and human activities could provide projections of the
dynamics of transboundary river cooperation and conflict. The results of both sensitivity
analysis and scenario analysis will provide useful insights for transboundary river management
in the future and can help the riparian countries to enter into regional cooperative behavior to
maximize collective benefits synergistically and advance water resource sustainability.
Finally, the kinds of transboundary dynamics that transpired in the Lancang-Mekong river basin
and described in the socio-hydrologic model are commonly found in many transboundary river
basins. In particular, benefit losses to downstream countries by the actions of upstream
countries such as dam construction, water extraction and pollution, can be counterbalanced by
the willingness to cooperate by upstream countries, by sharing some economic benefits with



downstream countries as compensation for their loss of economic benefits, in return from
indirect geopolitical benefits and investment opportunities. By capturing these mechanisms and
by accounting for the effects of hydrologic variability and reservoir releases on the economic
benefits of the various water uses in the quantification of willingness to cooperate, the socio-
hydrological model presented in this paper provides an objective scientific framework to
underpin transboundary water management and negotiations elsewhere.

**Code/Data availability**
The data is available on request from the corresponding author (tianfq@mail.tsinghua.edu.cn).

**Author contribution**
You Lu, Fuqiang Tian, Liying Guo, Iolanda Borzi and Rupesh Patil discussed the framework
of model. You Lu developed the model code and performed the simulations. Jing Wei,
Dengfeng Liu, Yongping Wei and David J. Yu discussed and revised the model. You Lu,
Fuqiang Tian and Murugesu Sivapalan prepared the manuscript with contributions from all co-
authors.

**Competing interests**
The authors declare that they have no conflict of interest.

**Acknowledgements**



We would like to acknowledge the Ministry of Science and Technology, China
(2016YFA0601603) and National Natural Science Foundation of China (51961125204) for the
financial support. We also acknowledge the support from the 2019 Summer Institute on Socio-
hydrology and Transboundary Rivers held in Yunnan University, China.

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





**List of Figure Captions**

Figure 1. Map of Lancang-Mekong River Basin, Subbasin Division and Hydrological Stations

Figure 2. Timeline of hydrological and anthropogenic events in Lancang-Mekong River Basin

Figure 3. Framework of Transboundary River Socio-Hydrological Model

Figure 4. Framework of simplified water system in Lancang-Mekong River Basin

Figure 5. Daily Runoff simulations at Jinghong (a), Chiang Saen (b), Luang Prabang (c) and

Pakse (d)

Figure 6. Reservoir storage and water release simulations of Xiaowan, Nuozhadu and Laos

Reservoirs in 2015 (a-f) and 2017 (g-i).

Figure 7. Water release of Xiaowan, Nuozhadu and Laos Reservoirs in non-flood seasons in

2015 (dry year) and 2017 (norm year) under different scenarios

Figure 8. Simulated storage dynamics of Xiaowan (a), Nuozhadu (b) and Laos reservoirs (c)

Figure 9. Benefit of upstream China (a) and Laos (b) under the three different scenarios

Figure 10. Agriculture and fishery benefit of downstream Thailand, Cambodia and Vietnam

under the three different scenarios

Figure 11. Simulation of cooperation demand of downstream and cooperation level of China

and Laos

Figure 12. Simulation of cooperation demand and newspaper sentiment analysis of Thailand


991                  Table 1. Reservoir information of Xiaowan and Nuozhadu

| Reservoir | Commissioned Year | Total Reservoir Storage (MCM) | Flood Limited Storage (MCM) | Dead Reservoir Storage (MCM) |
|---|---|---|---|---|
| Xiaowan | 2010 | 15,300 | 13,104 | 5,946 |
| Nuozhadu | 2012 | 21,749 | 19,344 | 10,414 |


993             Table 2.Irrigated agriculture information of Thailand, Cambodia and Vietnam

| | Thailand | Cambodia | Vietnam | Data Source |
|---|---|---|---|---|
| Rice price (USD/ton) | 243.8 | 267.6 | 248.0 | MRC (2018) |
| Irrigated Area (million ha) | 1.425 | 0.505 | 1.921 | Cramb (2020) |
| Rice yield (ton/ha) | 3.78 | 4.38 | 5.72 | MRC (2018) |
| Irrigation withdrawal (MCM) | 16240 | 1680 | 29120 | AQUASTAT |


995           Table 3. Simulated economic benefits in 2018 and statistical benefits

| Unit: Million USD | Simulated benefit | Benefit from statistical data |
|---|---|---|
| China hydropower | 1,767 | 2,000 |
| Laos hydropower | 1,083 | 1,076 |
| Thailand agriculture | 1,355 | 1,314 |
| Thailand fishery | 116 | 120 |
| Cambodia agriculture | 595 | 592 |
| Cambodia fishery | 1,146 | 1,188 |
| Vietnam agriculture | 2,011 | 2,727 |



| Vietnam fishery | 178 | 195 |
|---|---|---|






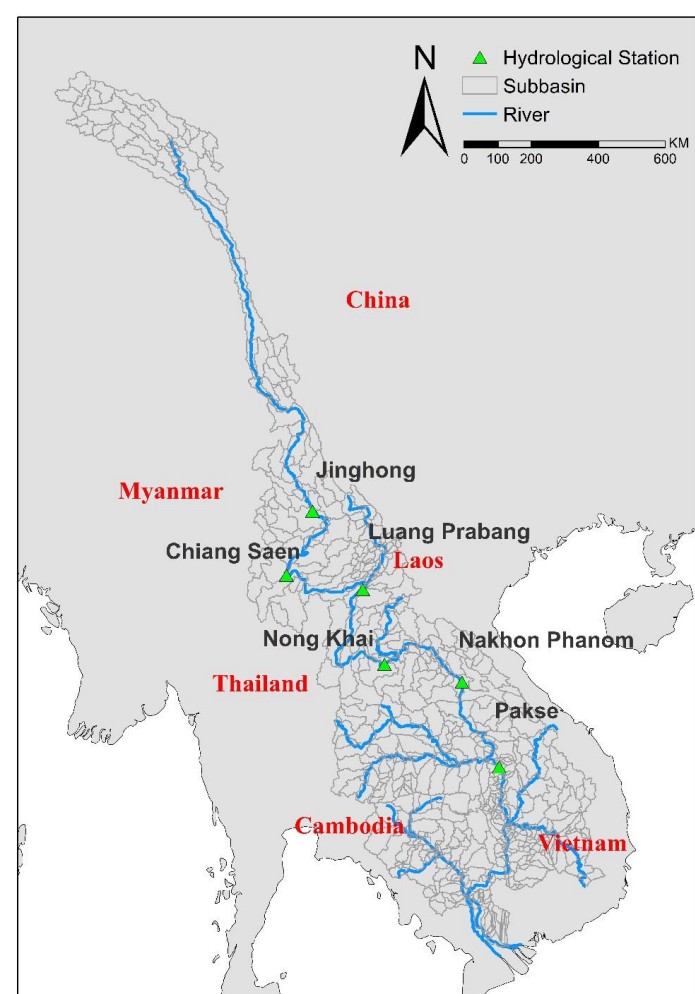


Figure 1. Map of Lancang-Mekong River Basin, Subbasin Division and Hydrological Stations





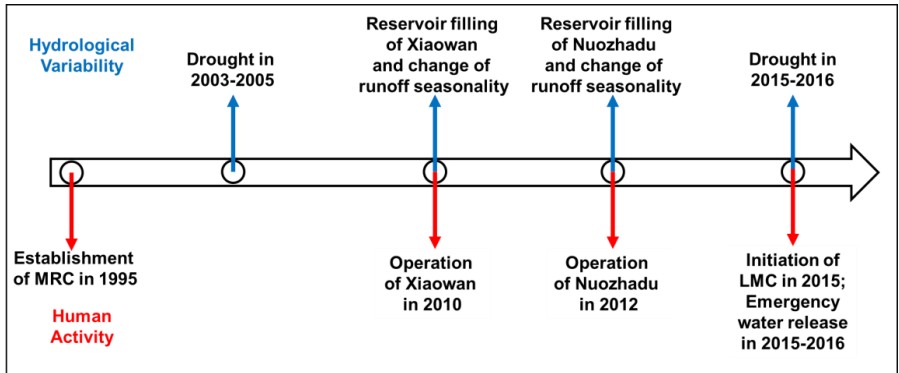


Figure 2. Timeline of hydrological and anthropogenic events in Lancang-Mekong River Basin





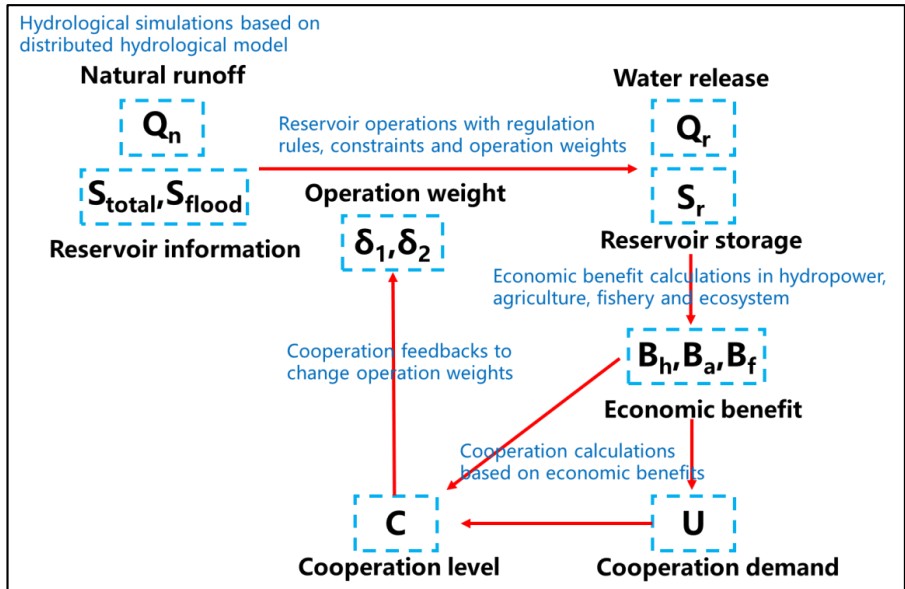

Figure 3. Framework of Transboundary River Socio-Hydrological Model





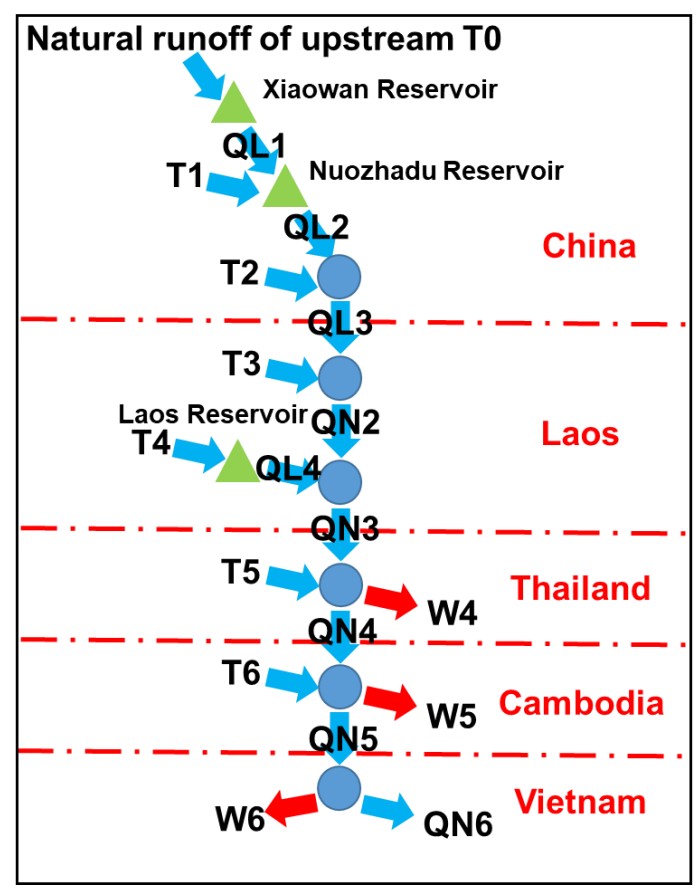


1007        Figure 4. Framework of simplified water system in Lancang-Mekong River Basin




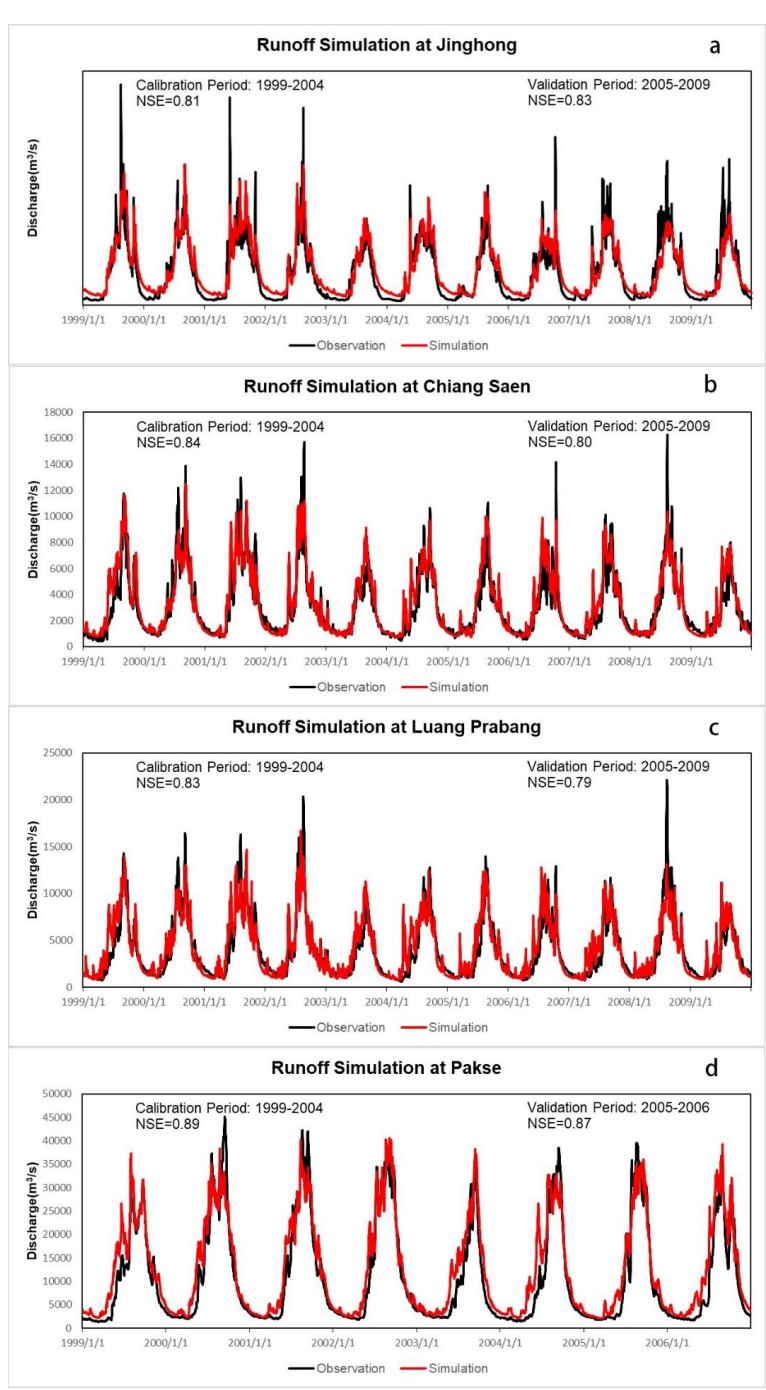


Figure 5. Daily Runoff simulations at Jinghong (a), Chiang Saen (b), Luang Prabang (c) and
Pakse (d)

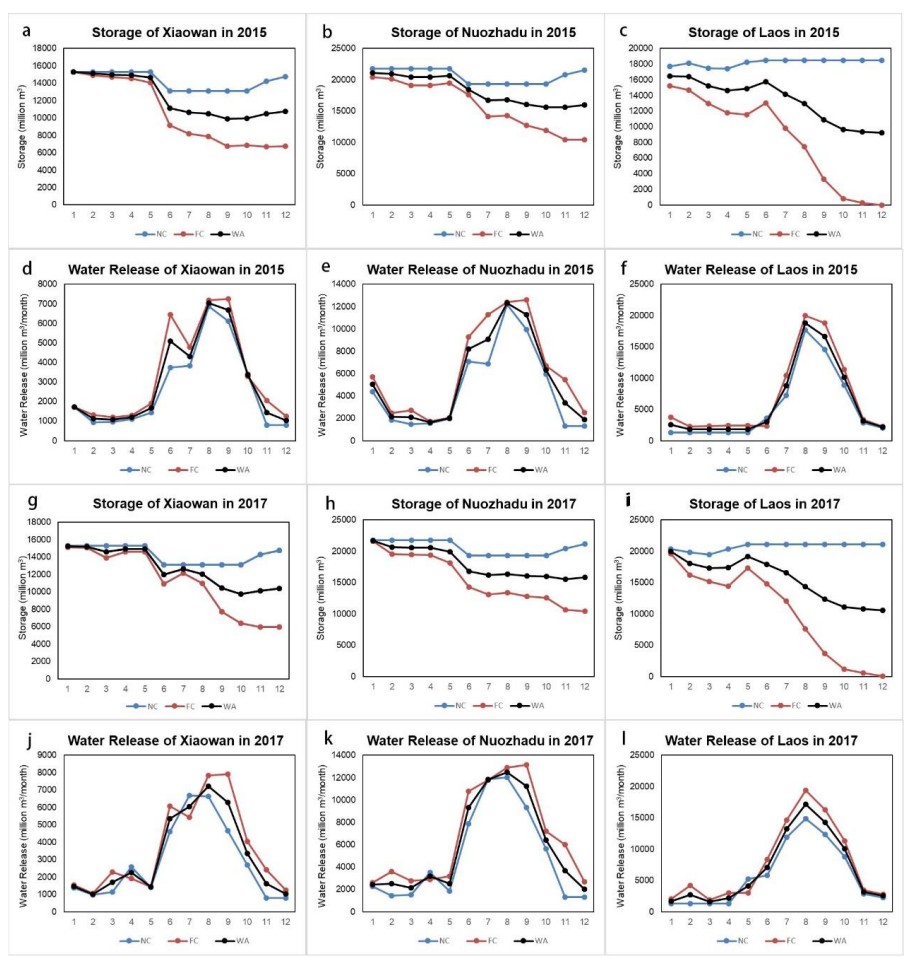


Figure 6. Reservoir storage and water release simulations of Xiaowan, Nuozhadu and Laos


Reservoirs in 2015 (a-f) and 2017 (g-i).







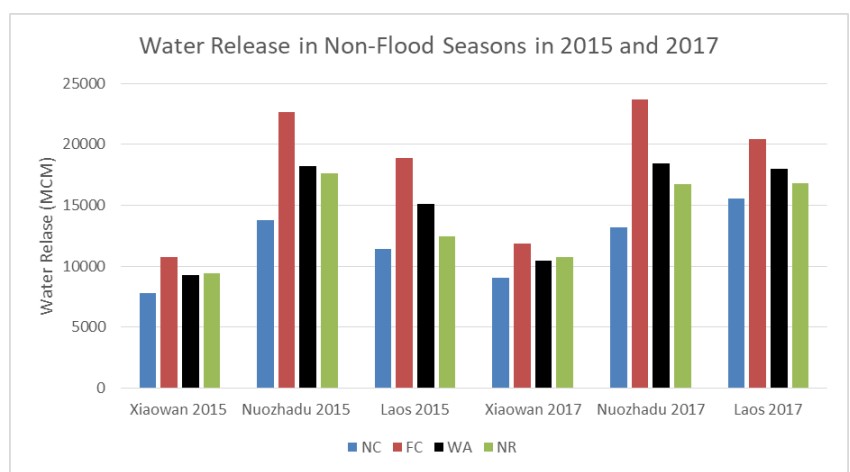


Figure 7. Water release of Xiaowan, Nuozhadu and Laos Reservoirs in non-flood seasons in
2015 (dry year) and 2017 (norm year) under different scenarios



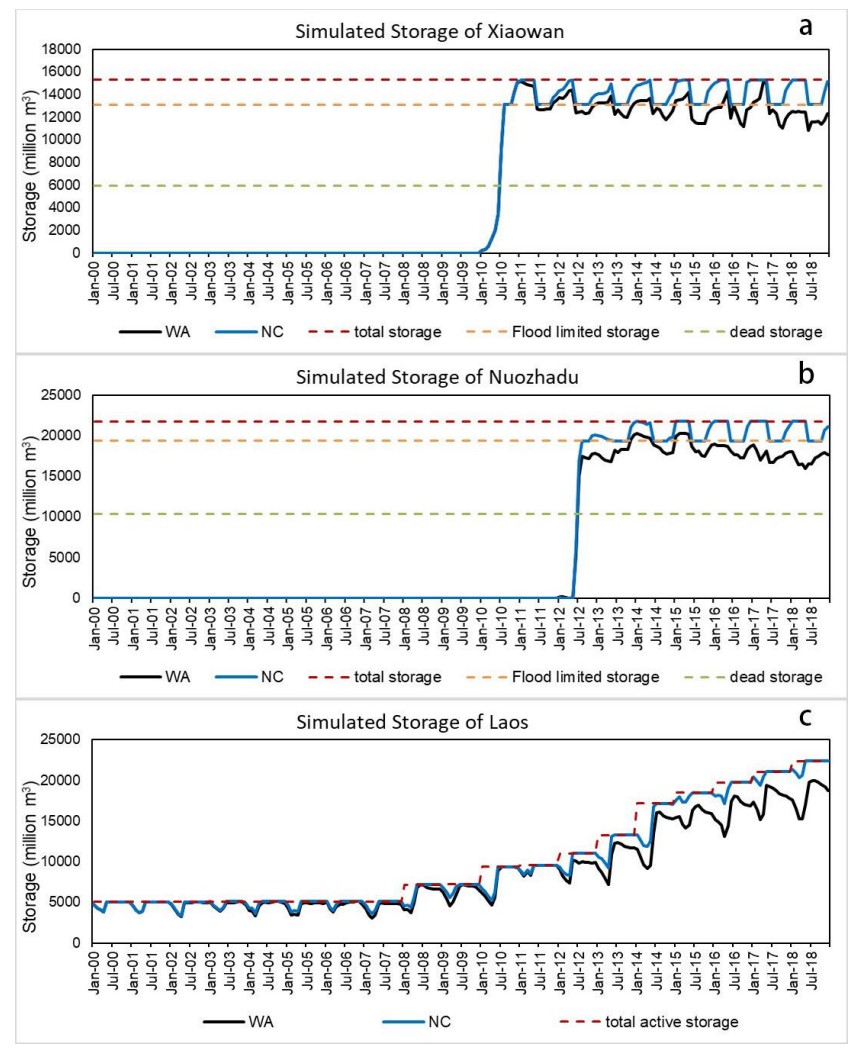


Figure 8. Simulated storage dynamics of Xiaowan (a), Nuozhadu (b) and Laos reservoirs (c)





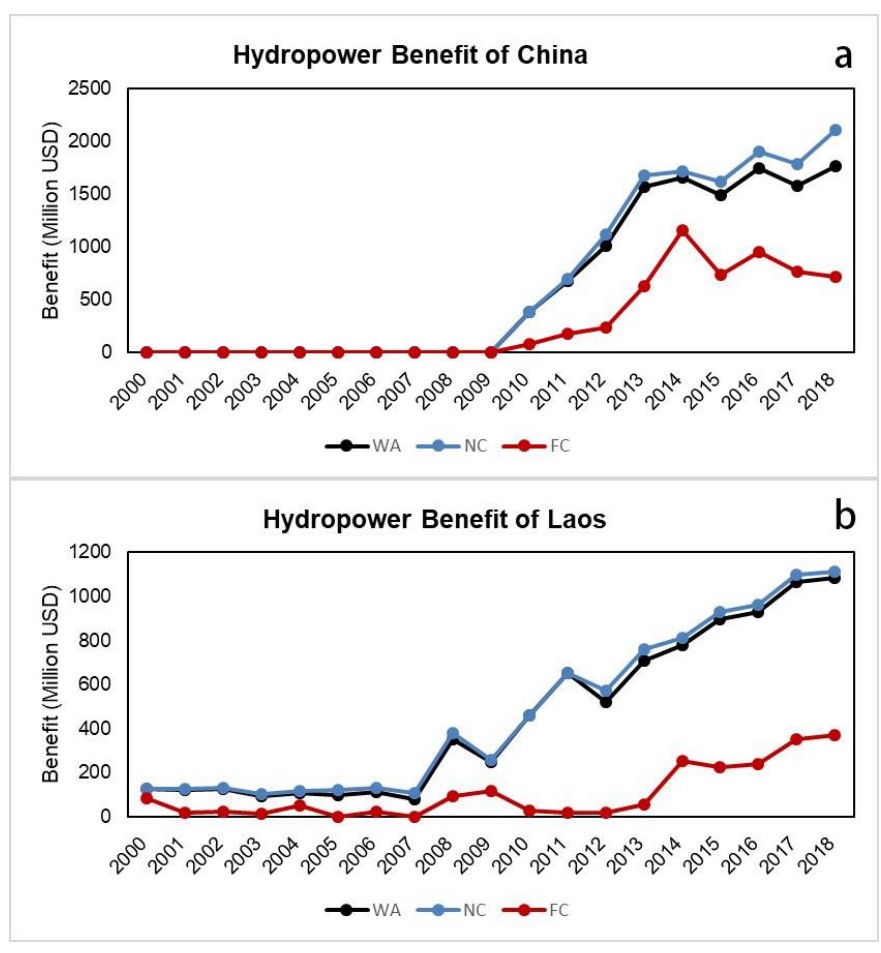


Figure 9. Benefit of upstream China (a) and Laos (b) under the three different scenarios



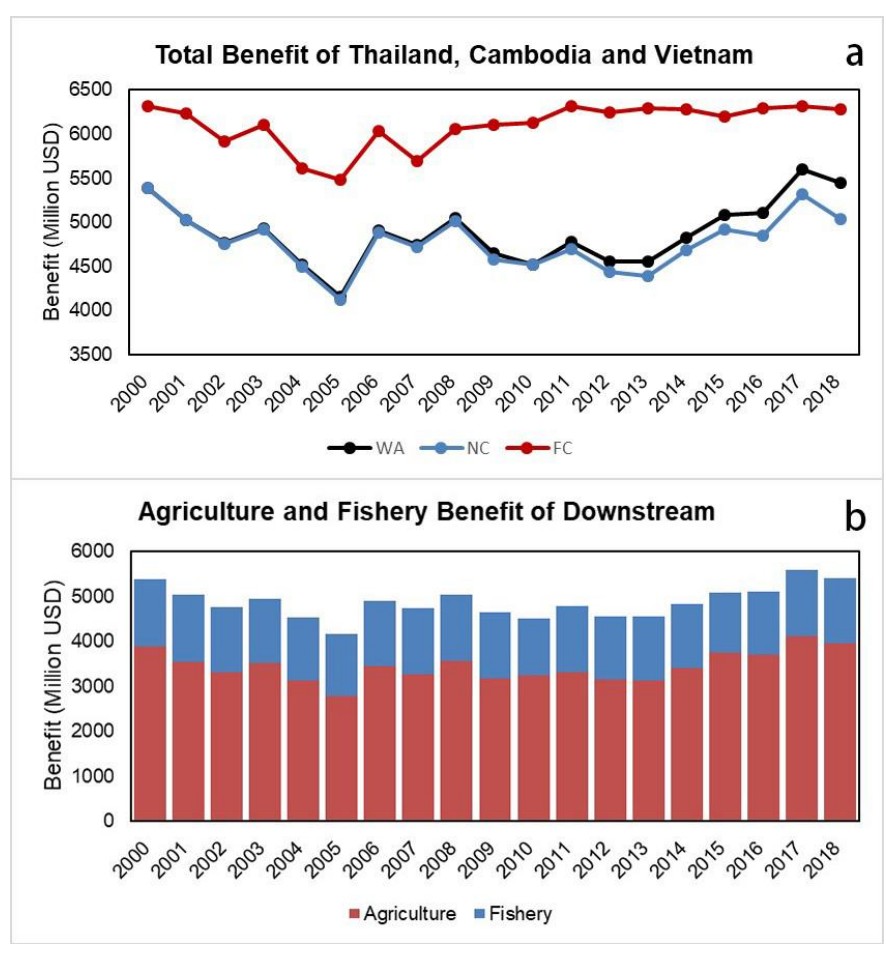


Figure 10. Agriculture and fishery benefit of downstream Thailand, Cambodia and Vietnam
under the three different scenarios





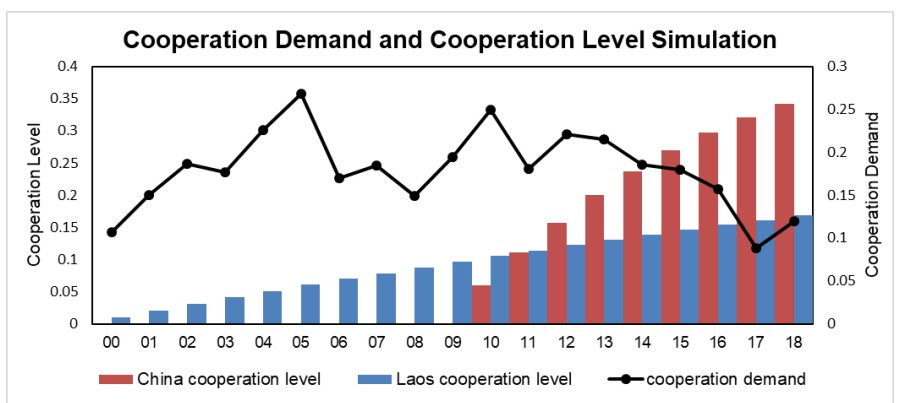


Figure 11. Simulation of cooperation demand of downstream and cooperation level of China
and Laos





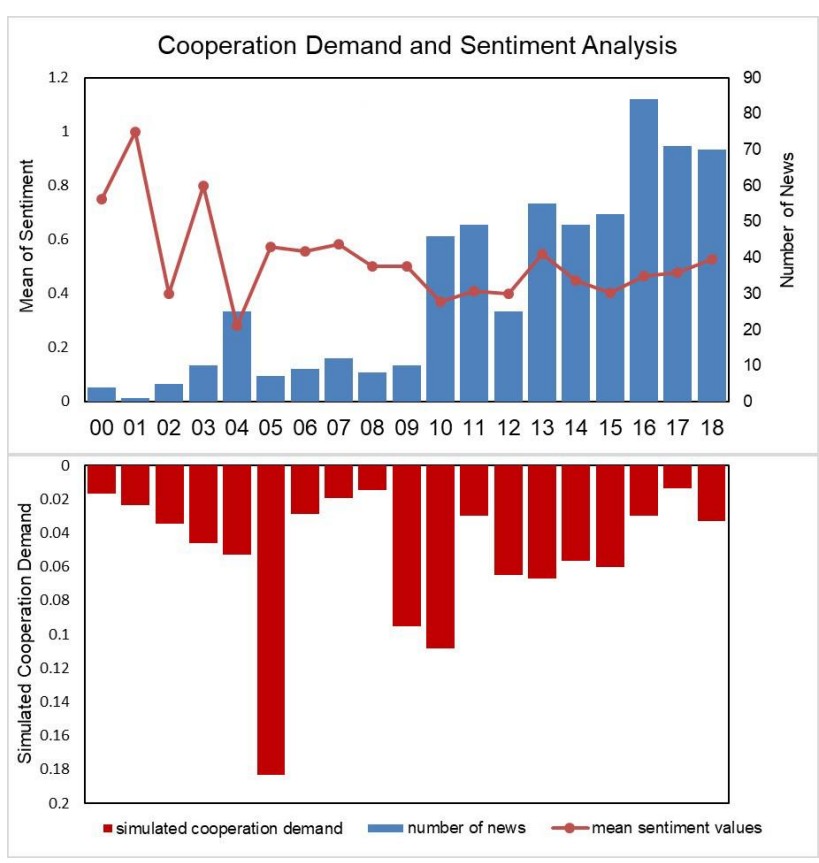


Figure 12. Simulation of cooperation demand and newspaper sentiment analysis of Thailand