# Peer review of "Socio-Hydrologic Modeling of the Dynamics of Cooperation in the Transboundary"

_Hydrology and Earth System Sciences, 2020_

## Referee Comment (RC1) · Gopal Penny (Referee) · 25 Sep 2020

Lu et al., 2020. HESSD.

General comments

The authors present a socio-hydrological model of cooperation in transboundary watersheds, using the Lancang-Mekong river basin as a case study. In the model (and case study), upstream countries seek to develop the river for hydropower generation whereas downstream countries rely on river flow for agriculture and fisheries. Cooperation in the model is realized when upstream riparian countries adjust reservoir
operations and forgo economic gains for the benefit of downstream riparian countries. This cooperation occurs in response to "cooperation demand" of downstream countries, which increases when ecosystem services decrease in these countries. The essential feedback within the model (i.e., cooperation) also depends on the disposition of the upstream riparian to their downstream partners, broadly representing the geopolitical relationship between countries. Cleverly, reservoir operation varies between the ideal operation for upstream countries (optimize hydropower) and downstream countries (e.g., optimize agricultural production), which allows cooperation to be quantified as a value between 0 and 1.

The manuscript focuses on model development and validation rather than hypothesis testing. The novelty of the manuscript therefore rests primarily in the quantitative formulation of transboundary cooperation, including a novel and parsimonious representation of transboundary politics and decision making. This question is of great concern to understanding long-term streamflow trajectories in transboundary basins. The hydrological portion of the model is rigorously validated, and the economics portion of the model is based on established models. The cooperative and political components of the model are also based on published literature. However, given the novelty of this aspect of the model and importance to the nonlinearity / feedback mechanism within the model, the cooperative aspects merit additional consideration within the manuscript, as I describe in "Specific comments".

Overall the manuscript is clearly written and presents an important and novel contribution to better understand and model cooperation in transboundary basins. My primary concern relates to parameter selection and calibration, especially those parameters pertaining to cooperation (as I describe below).

Specific comments

Although the cooperation portion of the model appears to be designed in a manner that is consistent with cooperation in the Mekong, parameterization and validation of

this component of the model is given relatively little attention. This raises the question of whether or not the (geo)politics represented in the model (specifically, variables P, s, and B_h,FC) can be parameterized a priori, or if they need to be calibrated a posteriori to cooperation. The method of selecting these variables should be more clearly described with within the manuscript (beyond a statement of their values on lines 609-613). This information is essential to understand how the model could be used to understand transboundary cooperation beyond the Mekong, and along those lines it would be helpful for the authors to more clearly detail how they envision future applications of the model would support (and help understand) transboundary cooperation in other regions. This would also give more weight to the statement (abstract, lines 42-44) that "the socio-hydrological model used here provides a useful new framework to investigate and improve transboundary water management elsewhere."

The dynamics of cooperation within the Mekong have been described qualitatively in other papers referenced in the manuscript. Indeed, these papers give confidence to the model as the authors demonstrate that the model is consistent with this qualitative narrative, both in terms of formulation and outcome. However, this raises the question of how the model generates additional insights about cooperation in the Mekong basin beyond what has already been described elsewhere. This particular aspect should be clarified within the manuscript — i.e., what specific insights about transboundary cooperation has the model generated that were not apparent from previous research?

An additional aspect of the manuscript that was novel was the use of Lexis-Nexis sentiment analysis, and I suggest this aspect be given more attention in the introduction (at present, it seems downplayed).

Technical corrections

The second sentence of the abstract seems informal and lacks precision, and I suggest replacing "etc." with either additional examples or more precise phrasing.

Parameter selection would be more appropriate in the model development section,

opposed to the results section (e.g., lines 609-613).

Lines 625 - 638 would fit better in the Model section, perhaps as a validation subsection.

Subsections in Section 2 are numbered incorrectly as 3.1, 3.2, etc (should be 2.1, 2.2, etc).

Using "iff" as a variable name is confusing, because in mathematics notation it means "if and only if" (line 402)

On the first read it was a bit confusing to have two equivalent variables for cooperation, delta2 and C. Perhaps it would be helpful to include a note in figure 3 that the two are equal.

---

## Referee Comment (RC2) · Anonymous Referee #2 · 8 Nov 2020

This paper presents a socio-hydrological model to analyze different levels of cooperation in the Mekong River basin. From two extremes scenarios – cooperative versus unilateral management – the negotiation space is determined and cooperation demands assessed. The topic is relevant to both scientists and policy makers working on the management of transboundary water resources. The paper is well organized and easy to follow.

In its present form, the paper is not ready for publication for the following reasons: (1) The literature review on the modeling and analysis of transboundary river basins is incomplete. I miss a description of the work done with multi-agent simulation models

(MAS) or with decentralized hydro-economic models, see e.g. Teasley, 2011 JWRPM; Giuliani, 2013 WRR; Jeuland et al., 2014. Explaining the differences between the proposed socio-hydrological model and those alternatives would enhance the scope of the manuscript. Right now, the novelty of the proposed modeling approach is not clearly established. (2) The authors should focus on the Lexis-Nexis sentiment analysis and how it can be used to construct scenarios or to "calibrate" a model of a coupled human-natural system. In my opinion, this is where the novelty lies. Shifting the focus on the Lexis-Nexis sentiment analysis however requires major rewriting that may be beyond a simple revision but I am convinced that it would definitely appeal to a broader audience. In that case, the rather coarse socio-hydrological model (see below) would then be used to support the Lexis-Nexis sentiment analysis. (3) The description of the socio-hydrological model should be improved. It is not clear why reservoirs are aggregated. Nor do we know how results are disaggregated. Moreover, the allocation rules, including reservoir operating rules, are barely described even though they play a critical role in the cooperative management of the river basin. To what extent can the operating rules be adjusted to accommodate downstream water demands? Is hedging considered? In case of water shortages, how is rationing implemented between water users/economic sectors/countries?

---

## Referee Comment (RC3) · Anonymous Referee #3 · 23 Nov 2020

I found this to be an interesting paper on an important topic. I have two major comments for consideration by the authors and editors.

1. While I find the sentiment analysis an interesting and potentially valuable aspect of the paper, it seems problematic to review only English language media. It has limited the authors to the study of a single downstream country, which in its own right is troubling when the purpose of the study is to contribute to cooperation analysis in a diverse, multi-country transboundary basin. Additionally, English language media is, in its very nature, designed for consumption by outsiders and elites. This could yield biased sentiment relative to native language publications, which may be more indicative

of populist sentiments and of the concerns and insecurities of the country.

2. I appreciate that the authors have considered some examples of specific drought events in their analysis, as it is an acknowledgement of the fact that cooperation can be most important for extreme events. However, I still worry that the approach presented here has a "look where the streetlight is" approach to assessing benefits and risks of cooperation vs. non-cooperation. For example, I see no consideration of uncertainty in the presentation of results, when in fact (as the authors acknowledge at times) there are deep uncertainties in many parameter estimates. For me this is clearest in the case of fisheries, where the potential for catastrophic impacts during extreme events or long-term ecological consequences of changing water conditions falls outside the range of any analysis that might be calibrated to the first few years of experience. Given the centrality of fishing in this basin, and the fact that it is one of the most contentious topic areas with respect to dam construction, I worry that an analysis like this one can understate the impact of dams and miss the low-probability, high-impact risks and benefits of changes in river management. It seems a problem more suited to robust decision making frameworks than to optimization. I don't insist that the authors completely change their framework for this manuscript. But at a minimum I would like to see: (1) acknowledgement of deep uncertainty and discussion of the implications of this uncertainty when interpreting results; (2) some form of sensitivity analysis that accounts for parameter uncertainty in the presentation of key results.

---

## Author Comment (AC1) · 23 Nov 2020

Reviewer#1:

Reviewer comment: General comments The authors present a socio-hydrological model of cooperation in transboundary watersheds, using the Lancang-Mekong river basin as a case study. In the model (and case study), upstream countries seek to develop the river for hydropower generation whereas downstream countries rely on river flow for agriculture and fisheries. Cooperation in the model is realized when upstream riparian countries adjust reservoir operations and forgo economic gains for the benefit of downstream riparian countries. This cooperation occurs in response to "cooperation demand" of downstream countries, which increases when ecosystem services decrease in these countries. The essential feedback within the model (i.e., cooperation) also depends on the disposition of the upstream riparian to their downstream partners, broadly representing the geopolitical relationship between countries. Cleverly, reservoir operation varies between the ideal operation for upstream countries (optimize hydropower) and downstream countries (e.g., optimize agricultural production), which allows cooperation to be quantified as a value between 0 and 1. The manuscript focuses on model development and validation rather than hypothesis testing. The novelty of the manuscript therefore rests primarily in the quantitative formulation of transboundary cooperation, including a novel and parsimonious representation of transboundary politics and decision making. This question is of great concern to understanding long-term streamflow trajectories in transboundary basins. The hydrological portion of the model is rigorously validated, and the economics portion of the model is based on established models. The cooperative and political components of the model are also based on published literature. However, given the novelty of this aspect of the model and importance to the nonlinearity / feedback mechanism within the model, the cooperative aspects merit additional consideration within the manuscript, as I describe in "Specific comments".

Author response: We would like to thank reviewer #1 for the constructive suggestions and comments, which we believe will help to improve the manuscript substantially. We agree to address the main issues raised by the reviewer, and our explanations and responses to all the reviewer's comments and questions are listed as follow.

Reviewer comment: Overall the manuscript is clearly written and presents an important and novel contribution to better understand and model cooperation in transboundary basins. My primary concern relates to parameter selection and calibration, especially those parameters pertaining to cooperation (as I describe below). Specific comments Although the cooperation portion of the model appears to be designed in a manner that is consistent with cooperation in the Mekong, parameterization and validation of

this component of the model is given relatively little attention. This raises the question of whether or not the (geo)politics represented in the model (specifically, variables P, s, and B_h,FC) can be parameterized a priori, or if they need to be calibrated a posteriori to cooperation. The method of selecting these variables should be more clearly described within the manuscript (beyond a statement of their values on lines 609- 613). This information is essential to understand how the model could be used to understand transboundary cooperation beyond the Mekong, and along those lines it would be helpful for the authors to more clearly detail how they envision future applications of the model would support (and help understand) transboundary cooperation in other regions. This would also give more weight to the statement (abstract, lines 42-44) that "the socio-hydrological model used here provides a useful new framework to investigate and improve transboundary water management elsewhere."

Author response: We appreciate this suggestion on the parameterization and validation of this model, which bring us to realize that the present introduction of parameterization is insufficient. Generally, the model aims to simulate the cooperation evolution in Lancang-Mekong River, which is a typical transboundary river where the upstream benefit from hydropower generation and the downstream gains irrigation and fishery products. Therefore, the model framework can be extended to many other transboundary rivers with dams in the upstream and agriculture and/or fishery in the downstream, when the input data and parameters are well adjusted. Critical parameters in the model are classified into two groups, i.e., parameters in benefit calculation module and parameters in policy feedback module. The parameters in benefit calculation module are extracted from literature or calibrated against the statistical benefits, which will be explained specifically in the next paragraph. The parameters in policy feedback module are set according to the reality of Lancang-Mekong cooperation and then adjusted based on the simulations of cooperation demand and cooperation level, which is still rough because there are limited research and knowledge on the quantification of cooperation and political benefits. Furthermore, as one result of another related paper we are still working on, the long-term sensitivity analysis of the parameters in our model

show that the simulation results are more sensitive to parameters in benefit calculation module, including price of rice pa, price of fishery pf, electricity price ph, but not sensitive to parameters in policy feedback module, such as sensitivity of agriculture loss and fishery loss $\varepsilon\_a$ and $\varepsilon\_f$, responsive change rate s and the shape parameter $\beta$. Since the parameters in benefit calculation module are reliable, the uncertainty of simulation is limited and the simulations are credible. When the model is applied in other similar transboundary rivers, the parameters in benefit calculation module need to be extracted from literature such as price data, or calibrated against statistical data of sector benefits, which will ensure the reliability of benefit calculation parameters. As for the parameters in policy feedback module, such as political factor P, responsive change rate s, and sensitivity of agriculture loss and fishery loss $\varepsilon\_a$ and $\varepsilon\_f$, they should be "calibrated" so that the simulated cooperation demand and cooperation level are consistent with reality and sentiment analysis data. When the model is applied in enormous cases, these policy feedback parameters could be investigated to find some patterns, which could be then used to determine the corresponding parameters a priori when apply to a new case. Specifically, the parameterization in Lancang-Mekong River is explained as follows. In the benefit calculation module, the price of rice pa in line 382 and the price of fishery pf in line 403 are extracted from MRC (2018). In the policy feedback module, the sensitivity of agriculture loss and fishery loss $\varepsilon\_a$ and $\varepsilon\_f$ in line 432 are assigned equally as 0.5 in this study, indicating the same importance of the two sectors. The assignments of responsive change rate s and political factor P are introduced in lines 605-611, and we will move the introduction of these two parameters forward to Section 3.4. As for the variables B_(h,FC) and B_(h,NC) mentioned by the reviewer, they indicate the hydropower benefits of upstream countries under altruistic scenario and self-interested scenario respectively, which are explained in 449-453. They are both calculated based on the equation (2) in line 344, in which the electricity price ph is extracted from MRC (2018), and the hydropower generation efficiency $\eta$ is calibrated against the annual power generation data (Yu et al. 2019). The monthly release Q_r and water head difference $\Delta$h are calculated under altruistic

scenario and self-interested scenario respectively in reservoir operation module. We will add the illustrations above to the manuscript to make the introduction of parameterization clearer.

Reviewer comment: The dynamics of cooperation within the Mekong have been described qualitatively in other papers referenced in the manuscript. Indeed, these papers give confidence to the model as the authors demonstrate that the model is consistent with this qualitative narrative, both in terms of formulation and outcome. However, this raises the question of how the model generates additional insights about cooperation in the Mekong basin beyond what has already been described elsewhere. This particular aspect should be clarified within the manuscript, i.e., what specific insights about transboundary cooperation has the model generated that were not apparent from previous research? An additional aspect of the manuscript that was novel was the use of Lexis-Nexis sentiment analysis, and I suggest this aspect be given more attention in the introduction (at present, it seems downplayed).

Author response: we thank the question raised by the reviewer. As discussed in the introduction and discussion part, the model is the first one to include the evolutionary dynamics of cooperation driven by hydrological variability and human activity as an internal variable. It enables the mid- and long-term evolutionary analysis of transboundary river cooperation and its driven mechanism, which is the unique insight this model can offer. Besides, the model is also a useful tool to analyze the impacts of hydrological, economic and political factors on transboundary river cooperation and project the evolution in the future. We have gained some insights in future projection based on the model, which will be explained in another paper under preparation. For short, new results include that under RCP6.0 emission scenario, the risk of conflict will not increase significantly in the basin, the downstream irrigation benefit will increase by 30% and Laos hydropower generation will increase by 120%. Both irrigation expansion and runoff decrease will lead to higher cooperation demand, and when the runoff under RCP6.0 decrease by 30% and irrigation expand as planned, reservoir construction and

operation of upstream will exceed increasing weight on political benefit of upstream in reducing conflict risks. But in this paper, development of the model is the main work of this manuscript, which has its own novelty and significance. We would like to highlight the distinctions from other model research and new insights the model can offer in the introduction part. Also, we agree that the use of sentiment analysis is a novelty of this manuscript. But we focus on sentiment analysis in another paper submitted to the same special issue on HESS (https://hess.copernicus.org/preprints/hess-2020-390/hess-2020-390.pdf), and we would like to reinforce the introduction of the sentiment analysis rather than focus on it in this manuscript.

Reviewer comment: Technical corrections The second sentence of the abstract seems informal and lacks precision, and I suggest replacing "etc." with either additional examples or more precise phrasing.

Author response: We appreciate the suggestion. We will replace "etc." with navigation and ecological services.

Reviewer comment: Parameter selection would be more appropriate in the model development section, opposed to the results section (e.g., lines 609-613).

Author response: We appreciate the suggestion. We will move this part forward to Section 3.4.

Reviewer comment: Lines 625 - 638 would fit better in the Model section, perhaps as a validation subsection.

Author response: We appreciate the suggestion. We will introduce the sentiment analysis after Section 3.4 and remain the comparison between sentiment analysis result and simulated cooperation demand as it be.

Reviewer comment: Subsections in Section 2 are numbered incorrectly as 3.1, 3.2, etc (should be 2.1, 2.2, etc).

Author response: We appreciate the suggestion. We will correct them.

Reviewer comment: Using "iff" as a variable name is confusing, because in mathematics notation it means "if and only if" (line 402)

Author response: We appreciate the comment. We will change iff to other expression.

Reviewer comment: On the first read it was a bit confusing to have two equivalent variables for cooperation, delta2 and C. Perhaps it would be helpful to include a note in figure 3 that the two are equal.

Author response: We appreciate the comment. We will add this equation in figure 3.

Reference:

MRC (2018). Summary State of the Basin Report 2018, Mekong River Commission.

Yu Y., Zhao J., Li D., et al. (2019). Effects of Hydrologic Conditions and Reservoir Operation on Transboundary Cooperation in the Lancang–Mekong River Basin. Journal of Water Resources Planning and Management 145(6): 04019020.

---

## Author Comment (AC2) · 23 Nov 2020

Reviewer #2:

Reviewer comment: This paper presents a socio-hydrological model to analyze different levels of cooperation in the Mekong River basin. From two extremes scenarios – cooperative versus unilateral management – the negotiation space is determined and cooperation demands assessed. The topic is relevant to both scientists and policy makers working on the management of transboundary water resources. The paper is well organized and easy to follow.

[Figure]

Author response: Thank the Referee #2 for the confirmation of rationality of our research. Also, we believe that all the comments will help to improve the manuscript substantially. We will address them in a point-by-point manner below.

Reviewer comment: In its present form, the paper is not ready for publication for the following reasons: (1) The literature review on the modeling and analysis of transboundary river basins is incomplete. I miss a description of the work done with multi-agent simulation models (MAS) or with decentralized hydro-economic models, see e.g. Teasley, 2011 JWRPM; Giuliani, 2013 WRR; Jeuland et al., 2014. Explaining the differences between the proposed socio-hydrological model and those alternatives would enhance the scope of the manuscript. Right now, the novelty of the proposed modeling approach is not clearly established.

Author response: Thank you for the comment. For literature review, several extant hydro-economic models were reviewed in lines 105-109 in the introduction part, and the reviewer's comment reminds us to realize that the literature review on multi-agent simulation models is still insufficient. We would like to review more literature on the modeling and analysis of transboundary river basins, particularly the multi-agent simulation models applied in transboundary rivers. For the novelty, our model has distinctions from the extant models. The extant hydro-economic models regard the cooperation as static and external variable. Whether cooperate or not, or the extent of cooperation, is set as boundary conditions of these models. However, as mentioned in the manuscript, transboundary river cooperation is evolutionary driven by hydrological, economic and political factors. To the best of our knowledge, this model developed in this work is the first one to include the evolutionary transboundary river cooperation as an internal variable, and couple the driven processes including hydrological variability, dam construction, political benefits, etc. To attain the goal, we also proposed the novel quantification of cooperation level and political benefits. This novelty enables the model to analyze the mid- and long-term cooperation dynamics in transboundary rivers, which cannot be achieved by previous models. We will supplement more explanations focus-

ing on the novelty in the revised manuscript, and clarify the differences between this model and the extant ones.

Reviewer comment: (2) The authors should focus on the Lexis-Nexis sentiment analysis and how it can be used to construct scenarios or to "calibrate" a model of a coupled human-natural system. In my opinion, this is where the novelty lies. Shifting the focus on the Lexis-Nexis sentiment analysis however requires major rewriting that may be beyond a simple revision but I am convinced that it would definitely appeal to a broader audience. In that case, the rather coarse socio-hydrological model (see below) would then be used to support the Lexis-Nexis sentiment analysis.

Author response: We thank this comment. Indeed, the sentiment analysis to validate the simulation of cooperation demand is important, and it gives us deeper insights into the influence factors of transboundary river cooperation. However, development of the model is the main work of this manuscript, and we believe that the model itself has its novelty and advantage in understanding and simulating the cooperation dynamics and driven mechanisms. The sentiment analysis in our manuscript is used to prove the validity of the model, instead of being the main work of the manuscript. There is another paper focusing on the sentiment analysis in Lancang-Mekong River, which is titled"An Analysis of Conflict and Cooperation Dynamics over Water Events in the Lancang-Mekong River Basin" and also under review in the same special issue on HESS (https://hess.copernicus.org/preprints/hess-2020-390/hess-2020-390.pdf). We believe both the model and sentiment analysis could be an independent and important work. We will improve the introduction of sentiment analysis, and discuss more deeply on the results of sentiment analysis. Thanks all the same.

Reviewer comment: (3) The description of the socio-hydrological model should be improved. It is not clear why reservoirs are aggregated. Nor do we know how results are disaggregated.

Author response: We thank this comment. In our study, the reservoirs in upstream

[Figure]

China and Laos are simplified and aggregated into 3 reservoirs, because we need to couple the reservoir operation module with the economic calculation and policy feedback module, and a more decentralized model with more agents will face the problem of computation time. According to the reservoir operation rules shown in lines 348-363, the altruistic scenario (full-cooperation scenario) is calculated by maximize downstream benefits, while the weight of this scenario equals to cooperation level. The cooperation level is dynamic and the calculation step is one month, which requires that the calculation time of each step including the optimization processes cannot be too long. Besides, the simplification of the reservoirs is reasonable and acceptable. As mentioned in lines 314-316, Xiaowan and Nuozhadu reservoirs account for 90% of the storage in China. The storage of the proxy Laos reservoir equals to the aggregation of all Laos reservoirs, and its hydropower generation is calibrated against the statistical data of the sum of hydropower generations in Laos. We will modify the description part of the model to clarify the rationality, and supplement more information about reservoir operation module.

Reviewer comment: Moreover, the allocation rules, including reservoir operating rules, are barely described even though they play a critical role in the cooperative management of the river basin. To what extent can the operating rules be adjusted to accommodate downstream water demands? Is hedging considered? In case of water shortages, how is rationing implemented between water users/economic sectors/countries?

Author response: We thank this comment. The regulation operating rules are explained in lines 348-363. Generally, in order to quantify the cooperation level, we assign it as the weight of altruistic scenario in line 367, which is the extent to which the operating rules are adjusted to accommodate downstream water demands. The altruistic scenario is to purely maximize the total downstream benefits. Hedging is not considered in this study. When there is water shortage, water allocation between countries and sectors will be implemented to maximize the total benefits of downstream three countries, with the constraint of water release from upstream countries. We will revise the

model description part, and add the explanation above to our manuscript to make it clearer.

---

## Author Comment (AC3) · 28 Nov 2020

Reviewer #3:

Reviewer Comment: I found this to be an interesting paper on an important topic. I have two major comments for consideration by the authors and editors.

Author Response: Thank the Referee #3 for the comments, which we believe will help improve the manuscript substantially. We will respond to the comments in a point-by-point manner as follow.

Reviewer Comment: 1. While I find the sentiment analysis an interesting and poten-

tially valuable aspect of the paper, it seems problematic to review only English language media. It has limited the authors to the study of a single downstream country, which in its own right is troubling when the purpose of the study is to contribute to cooperation analysis in a diverse, multi-country transboundary basin. Additionally, English language media is, in its very nature, designed for consumption by outsiders and elites. This could yield biased sentiment relative to native language publications, which may be more indicative of populist sentiments and of the concerns and insecurities of the country.

Author Response: We admit that the sentiment analysis based on English newspaper could yield biases compared to local language media. Although English is not frequently used in the Mekong countries, the English media is accessible to international audience, and serves the international audience who are interested in the affairs of these countries. The English media is considered a reference to the government's foreign policy (Curtin, 2012). Therefore, the English news articles can somehow reflect national interests and political responses that riparian countries want to deliver to the international public (Wei et al., 2020) in spite of its possible bias. On the other hand, the sentiment analysis based on local language media is tough at this stage. Since the local languages are diverse and not international language, it is too difficult for us to read manually, while the sentiment analysis algorithm could only deal with English articles, automatic analysis also faces difficulty. Among the downstream three countries, Thailand published the largest number of English news articles on the dam constructions of upstream, which could offer us a consecutive data series with relatively low uncertainty due to limited data. The validation against sentiment analysis of Thailand English media is thus a reasonable and feasible method. We have to acknowledge that this is a kind of "look where the streetlight is" manner, while it still shed light on the results produced by the model. We will admit the shortage and biases of the method, and explain more the rationality at this stage in the revision. Thanks for the comment.

Reviewer Comment: 2. I appreciate that the authors have considered some examples

of specific drought events in their analysis, as it is an acknowledgement of the fact that cooperation can be most important for extreme events. However, I still worry that the approach presented here has a "look where the streetlight is" approach to assessing benefits and risks of cooperation vs. non-cooperation. For example, I see no consideration of uncertainty in the presentation of results, when in fact (as the authors acknowledge at times) there are deep uncertainties in many parameter estimates. For me this is clearest in the case of fisheries, where the potential for catastrophic impacts during extreme events or long-term ecological consequences of changing water conditions falls outside the range of any analysis that might be calibrated to the first few years of experience. Given the centrality of fishing in this basin, and the fact that it is one of the most contentious topic areas with respect to dam construction, I worry that an analysis like this one can understate the impact of dams and miss the low-probability, high-impact risks and benefits of changes in river management. It seems a problem more suited to robust decision making frameworks than to optimization. I don't insist that the authors completely change their framework for this manuscript. But at a minimum I would like to see: (1) acknowledgement of deep uncertainty and discussion of the implications of this uncertainty when interpreting results; (2) some form of sensitivity analysis that accounts for parameter uncertainty in the presentation of key results.

Author Response: Thank the reviewer for suggestions on the uncertainty problem. We admit that the structure of model and parameterization could cause uncertainty, including the fishery benefit calculations mentioned by the reviewer, which could face deep uncertainty because of the simplification. We will acknowledge this in the revision and discuss the uncertainty when interpreting results. For the sensitivity analysis, we have chosen several critical parameters to calculate their sensitivity index when conducting future scenario analysis. Results show that among the parameters, simulations are more sensitive to the prices of crop, fishery and hydropower, and irrigation efficiency coefficient, when compared to the parameters in the policy feedback module. In this case, we will conduct Monte Carlo test of the sensitive parameters to find out the un-
certainty of simulated cooperation level, cooperation demand, upstream/downstream benefits caused by parameterization, and add this part to the revision.

Reference: Curtin, M.: Chinese media and globalization, Chinese Journal of Communication, 5, 1-9, 2012. Wei, J., Wei, Y., et al. An Analysis of Conflict and Cooperation Dynamics over Water Events in the Lancang-Mekong River Basin, Hydrology and Earth System Sciences Discussion, 2020.
* * *

---

## Author Response (AR2)

We would like to thank all reviewers for their constructive suggestions. We reply to every comment as follow. For clarity, all comments are marked in blue and answers are marked in black.

Reviewer comment:

My primary concerns with the original manuscript were that (i) it was unclear how the authors envisioned translating this model to other transboundary basins, and (ii) the paper lacked an adequate description of parameter selection and calibration.

The first concern (i) has been addressed within the manuscript conclusions. The authors have mostly addressed the second concern (ii) pertaining to parameter selection and calibration by describing their process throughout Section 3 and adding a new table (Table 3) that shows the value of key parameters and the range used in the uncertainty analysis. There are a couple of minor points that should still be addressed, as I describe in the comments below.

Author response: We thank the comments from reviewer#2. We would like to reply to the further questions raised by reviewer#2 as follow.

1. The distinction between parameter selection versus parameter calibration for P is unclear given differences between the author response and the manuscript. The authors' response states:

"As for the parameters in policy feedback module, such as political factor P, responsive change rate s, and sensitivity of agriculture loss and fishery loss $\varepsilon a$ and $\varepsilon f$, they should be "calibrated" so that the simulated cooperation demand and cooperation level are consistent with reality and sentiment analysis data."

This appears to contradict the manuscript, which states (L484):

"The parameters in the policy feedback are defined a priori because there is limited research and knowledge at present on the quantification of cooperation and political benefits, which need further investigation."

Author response: We thank the reviewer#2 to point out the issue here. In this study, the parameters in policy feedback module including political factor, responsive change rate, etc., are assigned as introduced in the manuscript. The uncertainty analysis in Figure 12 shows that, although the parameters in policy feedback module could lead to uncertainty of simulated cooperation demand of downstream, the trend and fluctuation pattern of simulations are similar, and Figure 11 shows that the simulated cooperation demand are consistent with sentiment analysis results. Therefore, in this study, we do not need to "calibrate" parameters in policy feedback module after the initial assignment of parameter values. The statement in response that parameters should be "calibrated" means that when simulations with assigned parameters disagree with sentiment analysis results, parameters should be adjusted. We clarified the selections of parameters in the revised manuscript.

2. In the uncertainty analysis (Fig. 12), why is there no effect of the China Political Factor on cooperation demand? Perhaps this has to do with the logistic structure of the model for the change in cooperation level C, but nevertheless this counters the narrative that indirect political benefits to China (and therefore willingness to cooperate) relieved economic pressure (and therefore cooperation demand) in downstream countries. This should be clarified.

Author response: Figure 12 shows that with different values of China political factors, downstream cooperation demands vary slightly. According to the structure of model, higher China political factor results in higher cooperation level of China. Downstream countries will get higher benefits, and downstream cooperation demand will thus decrease. However, the logistic structure of calculation of cooperation level lead to subtle distinctions of downstream cooperation in the uncertainty analysis in figure 12. It does not mean that China political factor has no effect on downstream demand. When the values of China political factor differ more remarkably, simulations will illustrate the pattern that higher China political factor leads to lower downstream cooperation demand. We emphasized the aim and implication of uncertainty analysis in the revised manuscript, and clarified that the impacts of certain parameter on simulation should be investigated with larger range of values and more tests.